# A combined opposite targeting of p110δ PI3K and RhoA abrogates skin cancer

Niki Tzenaki[1], Lydia Xenou[1], Evangelia Goulielmaki[1], Anna Tsapara[1], Irene Voudouri[1], Angelika Antoniou[1], George Valianatos[1], Maria Tzardi[2], Eelco De Bree[3], Aikaterini Berdiaki[4], Antonios Makrigiannakis[4] & Evangelia A. Papakonstanti [1][✉]

Malignant melanoma is the most aggressive and deadly skin cancer with an increasing incidence worldwide whereas SCC is the second most common non-melanoma human skin cancer with limited treatment options. Here we show that the development and metastasis of melanoma and SCC cancers can be blocked by a combined opposite targeting of RhoA and p110δ PI3K. We found that a targeted induction of RhoA activity into tumours by deletion of p190RhoGAP-a potent inhibitor of RhoA GTPase-in tumour cells together with adoptive macrophages transfer from δ$^{D910A/D910A}$ mice in mice bearing tumours with active RhoA abrogated growth progression of melanoma and SCC tumours. The efficacy of this combined treatment is the same in tumours lacking activating mutations in *BRAF* and in tumours harbouring the most frequent BRAF(V600E) mutation. Furthermore, the efficiency of this combined treatment is associated with decreased ATX expression in tumour cells and tumour stroma bypassing a positive feedback expression of ATX induced by direct ATX pharmacological inactivation. Together, our findings highlight the importance of targeting cancer cells and macrophages for skin cancer therapy, emerge a reverse link between ATX and RhoA and illustrate the benefit of p110δ PI3K inhibition as a combinatorial regimen for the treatment of skin cancers.

[1] Department of Biochemistry, School of Medicine, University of Crete, Heraklion, Greece. [2] Department of Pathology, School of Medicine, University of Crete, University Hospital, Heraklion, Greece. [3] Department of Surgical Oncology, School of Medicine, University of Crete, University Hospital, Heraklion, Greece. [4] Department of Obstetrics and Gynaecology, School of Medicine, University of Crete, University Hospital, Heraklion, Greece. ✉email: epapak@uoc.gr

Skin cancer is the most common malignancy and includes melanoma and non-melanoma skin cancers. Malignant melanoma (MM) is not common cancer however, MM is an aggressive and deadly skin cancer[1] and despite the increase in effective systemic treatment options for advanced stage or metastatic MM during the last decade, still many patients die from this disease[2]. Squamous Cell Carcinoma (SCC) is a common non-melanoma skin cancer of the squamous cells, which make up the main part of the epidermis of the skin and is caused by the cumulative lifetime exposure to ultraviolet radiation of the sun[3,4]. The treatment options for locoregionally advanced or metastatic SCC remain limited[5].

MM has an immune-related pathogenesis and recently, the intratumoural inflammatory microenvironment was considered as a better predictor of survival than the histopathological staging[6,7] and as an important tool for potential manipulation and proper monitoring of therapies[6,7]. Tumour-associated macrophages (TAMs) were found to be the major immune constituent of the MM microenvironment and their levels compared with the other leucocytes contribute toward MM's prognosis[8,9] representing a poor indicator of patients' outcome. Furthermore, TAMs were found to compose cancer stroma not only in melanomas but also in non-melanoma skin cancers[10] and to significantly contribute to disease development[11]. Cutaneous SCC has been characterised as the most common malignancy in immunosuppressed patients such as patients with chronic lymphocytic leukaemia or solid organ transplant recipients[12–15], suggesting that cutaneous SCC is a highly immunogenic tumour. It is also of note that high levels of TAMs were found peritumorally and intratumorally in cutaneous SCC from transplant and non-transplant patients compared with normal skin[16] indicating a major role of TAMs in the development of SCC. Moreover, melanoma cells are known to express high levels of autotaxin (ATX) and its product lysophosphatidic acid (LPA)[17–21] and numerous studies have revealed a critical role for ATX and LPA in regulating a variety of pathophysiological processes including inflammation, fibrosis, tumour growth, metastasis and chemo-resistance[22–34]. LPA was found to contribute to the development and maintenance of tumour microenviroment (TME) whereas in some cancers, TME and especially TAMs have been characterised as main producers of ATX[35,36]. A wealth of evidence therefore suggests that combination approaches targeting both cancer cells and TAMs may be clinically beneficial[10,11,37,38]; however, it remains a challenge.

The pathogenesis of several solid human cancers including melanoma and non-melanoma cancers has been correlated with deregulation of the PI3K/Akt/mTOR pathways[39–43]. Furthermore, various evidence have suggested that some melanomas have intrinsic activation of the PI3K pathway[44] whereas the activated PI3K pathway has also been implicated in the development of resistance of melanoma tumours to BRAF/MEK inhibitors[45–48] and to immune checkpoint inhibitors[49,50]. Increased expression of mTOR and cyclin-dependent kinase 2 has also be found in cutaneous SCC, compared to its premalignant forms[51] and the PI3K/Akt/mTOR signalling pathway has been correlated with the development of resistance to EGFR inhibitors in head and neck SCC[52]. Drugs targeting BRAF V600 mutations and immune checkpoints have shown impressive initial therapeutic responses however, either not all patients responded or there were patients who developed resistance to these treatments[53–56]. PI3K inhibition was therefore attracted the interest in overcoming the therapeutic resistance in melanoma[57–60] and SCC[52,61] however, the results obtained were greatly variable and moreover a clinical trial using a mTORC1 inhibitor combined with an EGFR inhibitor did not show significant benefits to the patients[52] indicating the need to understand more precisely how the PI3K pathway is involved in skin cancers.

Signalling mechanisms operating downstream of tyrosine kinases and Ras are mediated by the class IA subset of PI3Ks which are heterodimers made up of a p85 regulatory subunit and a 110 kDa catalytic subunit (p110α, p110β or p110δ)[62–64]. p110α and p110β were found to be globally expressed[65–68] whereas p110δ is predominantly expressed in white blood cells[69,70]. High expression levels of p110δ were also found in some cancer cell lines and human tissues of non-leucocyte origin[71–73] and although the gene encoding the p110δ PI3K is rarely mutated in cancers[74–82] a promising role of p110δ has recently been emerged in solid tumours expressing high levels of non-mutated p110δ [72,73,83,84]. Downstream molecules in PI3K signalling includes regulators of small GTPases such as guanosine nucleotide exchange factors (GEFs) and GTPase-activating proteins (GAPs) and Ser/Thr kinases such as PDK1 and Akt/PKB[85–88]. Small GTPases are activated (become GTP-bound) by GEFs whereas the return from their active state to an inactive state (GDP-bound) is catalysed by GAPs[89]. On the other hand, the PI3K/Akt signalling pathway is regulated by phosphatases with the phosphatase and tensin homologue deleted on chromo- some 10 (PTEN) lipid phosphatase being the most extensively investigated[90]. Previous data have documented that the p110δ PI3K acts through a complex feedback mechanism in which the p110δ PI3K positively regulates the p190RhoGAP activity leading to negative regulation of RhoA GTPase and PTEN[91]. This mechanism revealed that PTEN is negatively controlled by p190RhoGAP[91].

In this work, we demonstrate that a combined treatment approach consisting of suppressed p190RhoGAP expression (leading to constantly increased activity of RhoA) into tumours and inactivation of p110δ PI3K in macrophages blocks melanoma and SCC progression. Our data show that the efficiency of this combined treatment is associated with decreased ATX expression in tumour cells and tumour stroma and emerge a reverse link between ATX and RhoA. Notably, the efficacy of this treatment approach is independent on BRAF mutational status of melanoma cells. We further show that although pharmacological inhibition of ATX activity *per os* is not efficient to prevent tumour growth because of a positive feedback regulation that induces the expression of ATX in tumour cells, the combined opposite targeting of p110δ and RhoA bypasses this positive feedback expression of ATX. Overall, this work paves the way for designing effective targeted therapies against skin cancer.

## Results

**Impact of combined inactivation of p110δ in macrophages with induced RhoA activity into tumours on melanoma tumour growth**. We first aimed to investigate the potential functional role of p110δ PI3K activity in macrophages in melanoma progression. We thus tested the impact of the targeted p110δ PI3K inactivation in macrophages on melanoma tumour growth by carrying out adoptive macrophage transfer experiments in B16 tumour-bearing NOD *scid* gamma (NSG) mice that have defective macrophages and lack T cells. When macrophages from δ[D910A/D910A] mice, which express genetically inactivated p110δ[92] were transferred into NSG mice the tumour burden was significantly reduced compared to that observed in mice that received macrophages from WT mice (Fig. 1a). In addition, immunostaining of tumour samples with the macrophage-specific antigen F4/80 showed that the abundance of macrophages into tumour sites was significantly reduced in mice receiving δ[D910A/D910A] macrophages compared with mice receiving WT macrophages (Fig. 1b). We also assessed the impact of oral administration of the p110δ selective inhibitor IC87114[93]

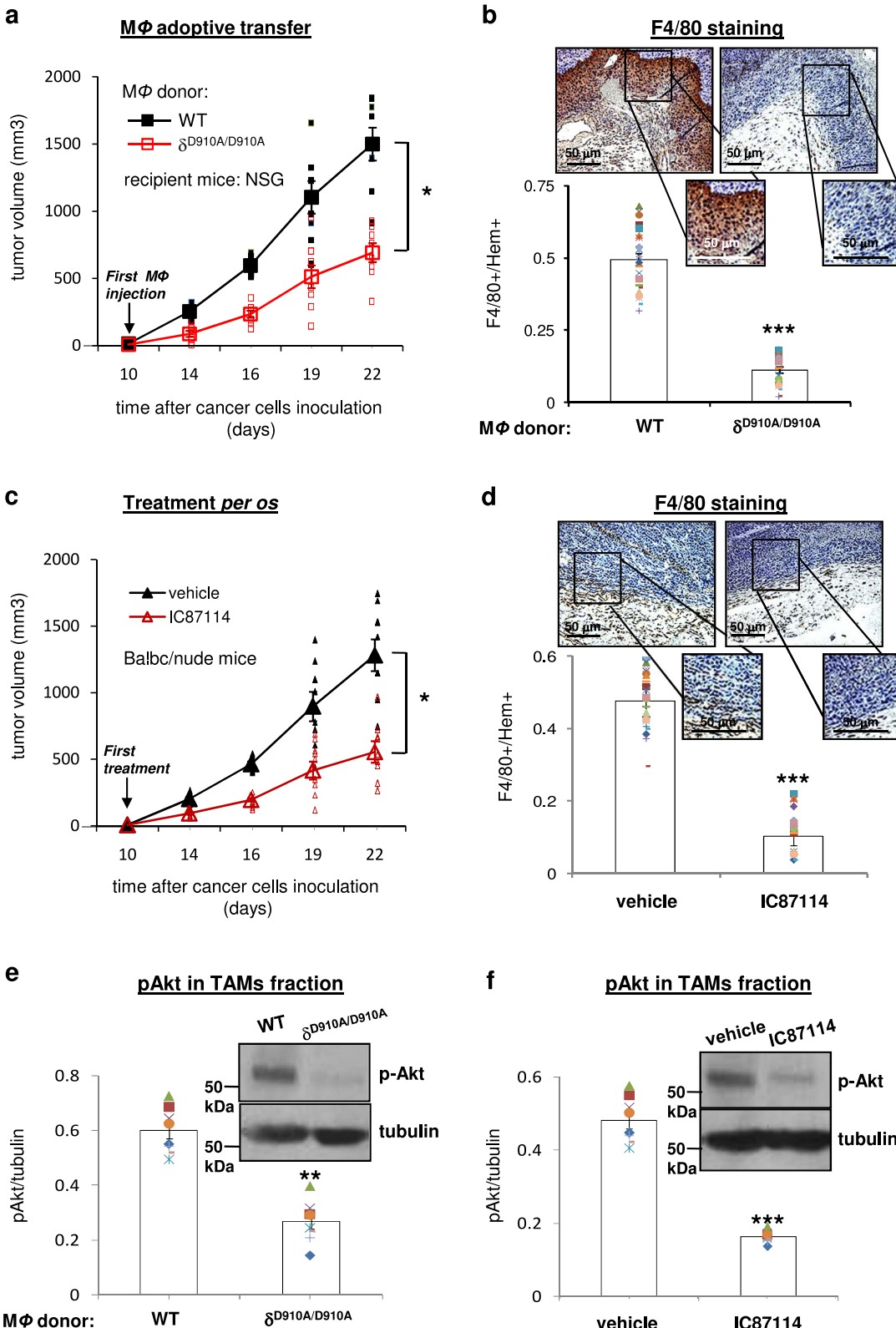

or vehicle on tumour growth and on the recruitment of macrophages to tumour sites in B16-bearing Balb/c nude mice which produce normal macrophages and lack T cells. Surprisingly, *per os* pharmacological inactivation of p110δ suppressed B16 tumour growth (Fig. 1c) to a similar extent as targeted inactivation of p110δ in macrophages (Fig. 1a). The abundance of macrophages was also drastically reduced in mice receiving per os IC87114

(Fig. 1d). To confirm the effect of p110δ inactivation on TAMs we isolated TAMs from excised tumours and tested their levels of Akt phosphorylation. We, indeed, found that the phosphorylation of Akt was decreased in TAMs fraction of tumours that were excised from mice receiving δ^D910A/D910A macrophages compared to that observed in mice receiving WT macrophages (Fig. 1e and Supplementary Fig. 8) as well as in TAMs fraction of

**Fig. 1 Targeted inactivation of p110δ PI3K in macrophages is sufficient to regress melanoma tumour growth. a** Impact of adoptive transfer of WT or δ$^{D910A/D910A}$ macrophages (from day +10) into NSG mice on B16 primary tumour growth. $n = 8$ mice/group. **b** Impact of adoptive transfer of WT or δ$^{D910A/D910A}$ macrophages into NSG mice on the recruitment of macrophages to tumour sites. Representative images of immunohistochemical staining with anti-F4/80 antibody (brown) and Hematoxylin (blue) in representative sections of tumours from NSG mice received WT or δ$^{D910A/D910A}$ macrophages Insets: magnified region delimited by grey rectangle in panel. Scale bar = 50 μm (upper panel). Comparison of F4/80-positive cells in tumours from NSG mice received WT or δ$^{D910A/D910A}$ macrophages (lower panel). **c** Growth of primary B16 tumours, inoculated in BALB/c nude mice. Mice were treated once daily *per os* with vehicle or IC87114 (35 mg/kg) from day +10. $n = 9$ mice/group. **d** Immunohistochemical staining of macrophage specific antigen F4/80 (brown) and Hematoxylin (blue) in representative sections of tumours excised from BALB/c nude mice that were treated *per os* with vehicle or IC87114 (35 mg/kg) Insets: magnified region delimited by grey rectangle in panel. Scale bar = 50 μm (upper panel). Comparison of F4/80-positive cells in tumours of IC87114-treated and vehicle-treated mice (lower panel). **e** Western blot analysis of Akt phosphorylation in TAMs fraction that was isolated from tumours that were excised from mice receiving WT or δ$^{D910A/D910A}$ macrophages. **f** Western blot analysis of Akt phosphorylation in TAMs fraction that was isolated from tumours that were excised from mice treated with IC87114 or vehicle. The bands of tubulin presented for normalisation in (**e**) and (**f**) were derived from different membranes but from the same cell lysates. Each symbol on the different groups denotes data from a different animal of the respective group. All graphs represent means ± s.e.m. Statistically significant differences are indicated by * ($P < 0.05$) or ** ($P < 0.01$), or *** ($P < 0.001$), as determined by the one-way ANOVA and the Mann-Whitney $U$ test.

tumours that were excised from mice treated with IC87114 compared to that found in mice that were treated with vehicle (Fig. 1f and Supplementary Fig. 8). These data indicate that the pharmacological inactivation of p110δ PI3K prevents melanoma tumour growth by a predominant targeting of macrophages without having a valuable effect on melanoma cells.

We next explored whether a combined targeted increase in activity of the PTEN tumour suppressor protein in melanoma cells could further regress melanoma tumour growth and metastasis. To assess this, we evaluated the impact of p190Rho-GAP deletion in tumour cells on tumour growth and metastasis. p190RhoGAP is a potent inhibitor of RhoA GTPase by catalysing the return of RhoA-GTP (active state) to RhoA-GDP (inactive state)[89] and therefore its deletion keeps RhoA active. On the other hand, it is known that RhoA mediates PTEN activation[91,94,95]. Indeed, intratumoural injection of p190RhoGAP siRNA suppressed the expression of p190RhoGAP as it was revealed by western blotting (Supplementary Fig. 1a, left panel and Supplementary Fig. 12a) and immunohistochemistry (Supplementary Fig. 1a, right panel) and, as it was expected, induced the levels of RhoA-GTP (Supplementary Figs. 1b and 12b). p190Rho-GAP functions predominantly on RhoA; however, it has been found to function as a GAP for Rac1 in cells[96]. Therefore, we tested the activation of Rac1 which found unaffected (Supplementary Fig. 1c and Supplementary Fig. 12c) indicating that p190RhoGAP affects selectively RhoA under these experimental conditions. Furthermore, intratumoural injection of p190Rho-GAP siRNA induced the activity of PTEN (Supplementary Fig. 1d) whereas suppressed the phosphorylation of Akt (Supplementary Fig. 1e and Supplementary Fig. 12e) in B16 melanoma tumours. Moreover, intratumoural administration of p190RhoGAP siRNA reduced tumour growth (Supplementary Fig. 2a) and proliferative rate of tumours (Supplementary Fig. 2b) whereas induced apoptosis in tumour cells (Supplementary Fig. 2c). Interestingly, when intratumoural p190RhoGAP siRNA injections in B16 tumours was combined with adoptive transfer of δ$^{D910A/D910A}$ macrophages into B16 tumour-bearing NSG mice there was an almost complete blockade of tumour growth compared to mice receiving WT macrophages alone or WT macrophages and p190RhoGAP siRNA injections (Fig. 2a). The massive decrease in tumour burden was also reflected to an extensive decrease in tumour mass in tumours harvested from mice receiving p190RhoGAP siRNA and δ$^{D910A/D910A}$ macrophages compared to mice receiving WT macrophages alone or WT macrophages and p190RhoGAP siRNA (Fig. 2b). The effectiveness of the combined targeting of RhoA in tumour cells and p110δ in macrophages was found to be the same either when interference with p110δ activity in macrophages starts concomitantly with (Fig. 2a), precedes (Fig. 2c) or follows (Fig. 2d) the induction of RhoA activity into tumours.

**The combined opposite targeting of p110δ and RhoA affects proliferation, apoptosis and metastasis of melanoma cells.** We then sought to determine the direct effect on tumour cells of the combined opposite targeting of p110δ PI3K and RhoA by investigating first whether this treatment affects the proliferation rate and apoptosis of tumour cells. The BrdU-positive cells (Fig. 3a) were significantly reduced in tumour specimens from mice receiving p190RhoGAP siRNA and δ$^{D910A/D910A}$ macrophages compared with mice receiving p190RhoGAP siRNA and WT macrophages, indicating that the proliferative rate of tumours was prevented whereas the number of TUNEL-positive cells was significantly increased in mice receiving p190RhoGAP siRNA and δ$^{D910A/D910A}$ macrophages (Fig. 3b), suggesting induced apoptosis in tumour cells. We then assessed the impact of the combined opposite targeting of p110δ PI3K and RhoA on cancer cells metastasis by determining the tumour cell blood burden and the expression of vimentin in the lungs. It is known that when the blood is collected from the right atrium of the heart before filtration by the lungs, the number of tumour cells in the blood is a direct evaluation of intravasation and therefore of efficiency of metastasis[97], whereas elevated expression of vimentin is correlated with lung invasion of cancer cells[98,99]. Under conditions of adoptive transfer of δ$^{D910A/D910A}$ macrophages in B16 tumour bearing NSG mice, the tumour blood burden was significantly reduced (Fig. 3c) indicating that the spontaneous intravasation and therefore metastasis is prevented when p110δ in macrophages is inactive. Furthermore, the expression of vimentin in the lungs was found to be significantly reduced in mice receiving p190RhoGAP siRNA and δ$^{D910A/D910A}$ macrophages compared with mice receiving p190RhoGAP siRNA and WT macrophages (Fig. 3d), reflecting decreased invasion of cancer cells. The functional importance of the combined opposite targeting of p110δ PI3K and RhoA in preventing the proliferation rate, apoptosis and metastasis of tumour cells was the same either when interference with p110δ activity in macrophages starts concomitantly with (Fig. 3), precedes (Supplementary Fig. 3), or follows (Supplementary Fig. 4) the induction of RhoA activity into tumours.

**The efficacy of the combined treatment against p110δ PI3K and p190RhoGAP is independent on BRAF mutational status of melanoma cells.** Given that in human melanomas at least 60% of tumours express *BRAF* mutations[100], we next explored whether the efficiency of the opposite targeting of p110δ PI3K and

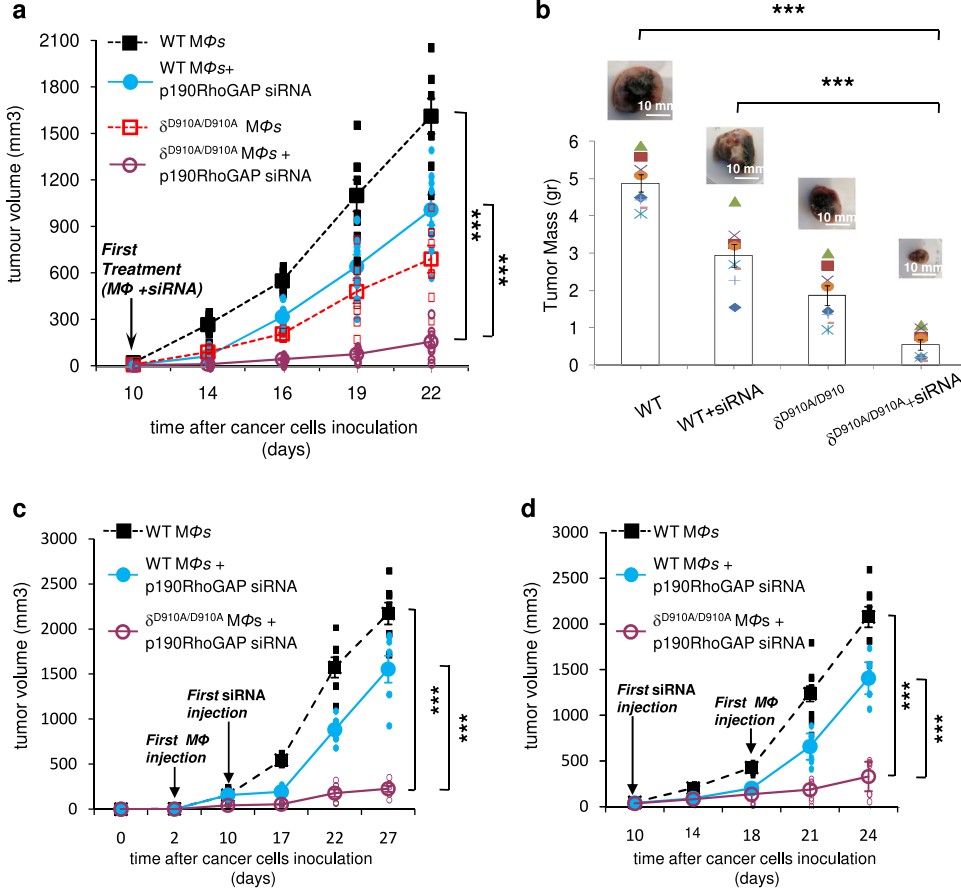

**Fig. 2 Combined inactivation of p110δ in macrophages with induced RhoA activity into tumours blocks melanoma tumour growth. a** NSG mice were inoculated with B16 cells on day 0 and treated with intravenous injections of WT macrophages or intravenous injections of WT macrophages and intratumoural injections of p190RhoGAP siRNA or intravenous injections of δ$^{D910A/D910A}$ macrophages or intravenous injections of δ$^{D910A/D910A}$ macrophages and intratumoural injections of p190RhoGAP siRNA on day +10 and on every other day until the end of the experiment. Tumour growth was measured. $n = 8$ mice/group. **b** Comparison of tumour mass of tumours excised from mice that were treated with intravenous injections of WT macrophages or intravenous injections of WT macrophages and intratumoural injections of p190RhoGAP siRNA or intravenous injections of δ$^{D910A/D910A}$ macrophages or with intravenous injections of δ$^{D910A/D910A}$ macrophages and intratumoural injections of p190RhoGAP siRNA. Scale bar = 10 mm. **c** Three groups of NSG mice were inoculated with B16 cells on day 0 and then two groups were treated with intravenous injections of WT and one group with δ$^{D910A/D910A}$ macrophages on day +2 and on every other day until the end of the experiment. The one group of mice that were received intravenous injections of WT macrophages and the group of mice that were received δ$^{D910A/D910A}$ macrophages were additionally treated with intratumoural injections of p190RhoGAP siRNA from day +10. All treatments were continued every other day until the end of the experiment and tumour growth was measured. $n = 7$ mice/group. **d** Three groups of NSG mice were inoculated with B16 cells on day 0 and then two groups of mice were treated with intratumoural injections of p190RhoGAP siRNA on day +10 and on every other day until the end of the experiment. Mice of those two groups were additionally treated with intravenous injections of WT or δ$^{D910A/D910A}$ macrophages respectively, from day +18 and on every other day until the end of the experiment. Mice of the third group, that were not received intratumoural injections of p190RhoGAP siRNA, were also treated with intravenous injections of WT macrophages from day +18 and on every other day until the end of the experiment. Tumour growth was measured. $n = 7$ mice/group. Each symbol on the different groups denotes data from a different animal of the respective group. All graphs represent means ± s.e.m. Statistically significant differences are indicated by * ($P < 0.05$) or ** ($P < 0.01$) or *** ($P < 0.001$), as determined by the one-way ANOVA and the Mann-Whitney $U$ test.

RhoA in preventing melanoma progression is altered when *BRAF* is mutated. B16 melanoma cells almost lack activating mutations in *BRAF*; however 451Lu human melanoma cells harbour the BRAF(V600E) which is the most frequent mutation in human melanoma[101]. We therefore assessed the response of 451Lu-tumour bearing NSG mice on the opposite targeting of p110δ PI3K and RhoA. The intratumoural injection of p190RhoGAP siRNA in 451Lu tumours suppressed the expression of p190RhoGAP (Supplementary Figs. 5a and 13a) and induced the activity of RhoA (Supplementary Figs. 5b and 13b) and PTEN (Supplementary Fig. 5c) whereas suppressed the phosphorylation of Akt (Supplementary Figs. 5d and 13d). The tumour burden was abolished in 451Lu-tumour-bearing mice receiving p190RhoGAP siRNA and δ$^{D910A/D910A}$ macrophages compared

to mice receiving WT macrophages alone or WT macrophages and p190RhoGAP siRNA (Fig. 4a). Furthermore, the tumour mass (Fig. 4b) as well as the number of tumour cells in the blood (Fig. 4c) were almost completely blocked in 451Lu-tumour bearing mice receiving p190RhoGAP siRNA and δ$^{D910A/D910A}$ macrophages confirming that the efficacy of this treatment approach is independent on BRAF mutational status of the tumour cells.

It is also noteworthy that in a limited number of PDTXs models that were developed by the implantation of human melanoma specimens, following surgical removal from patients tumour into an NSG mouse, the combined opposite targeting of p110δ PI3K and RhoA also abolished melanoma progression (Fig. 4d).

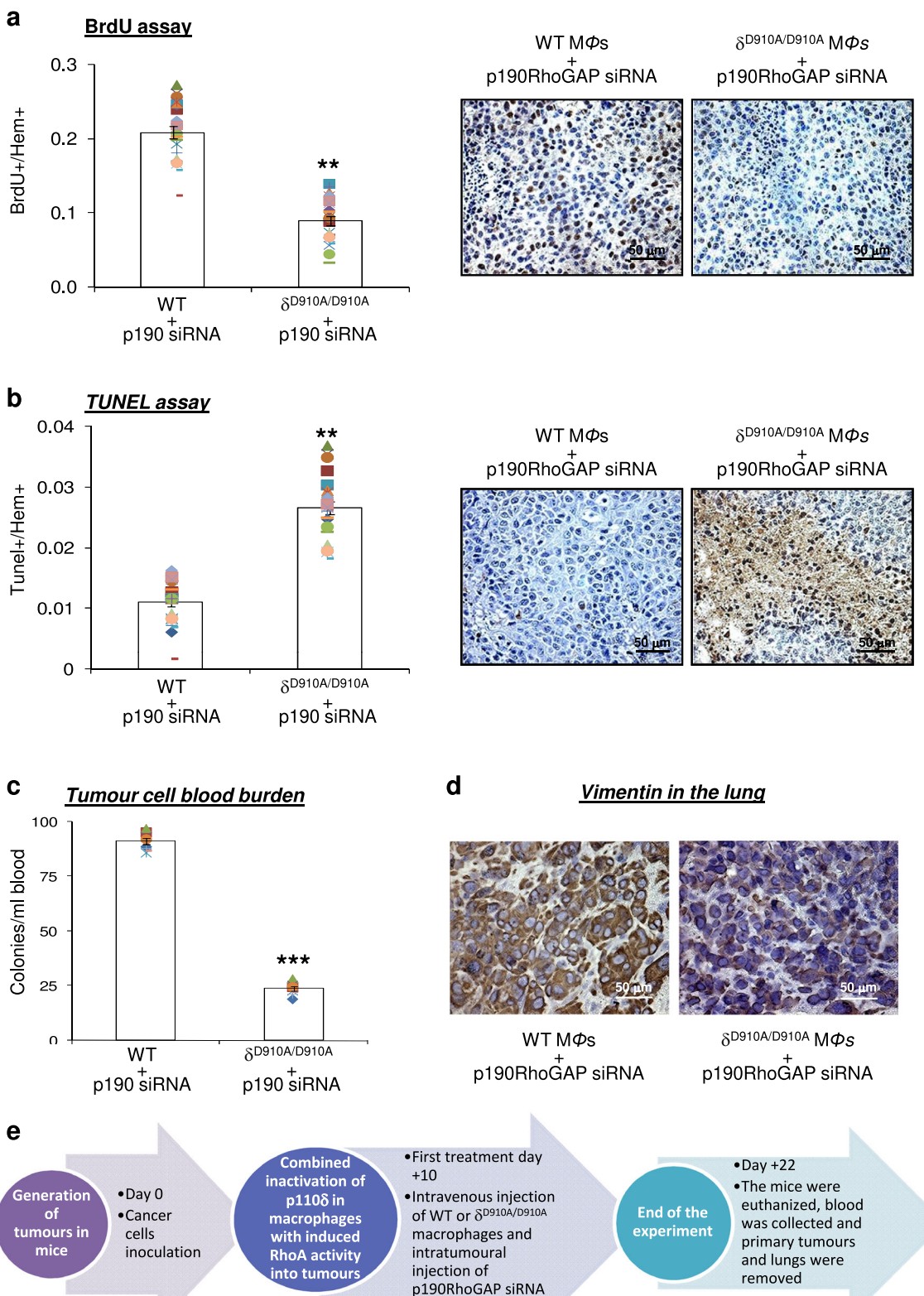

**Impact of combined inactivation of p110δ in macrophages with induced RhoA activity into tumours on SCC tumour growth.** We next explored the possible impact of the combined opposite targeting of p110δ PI3K and RhoA on SCC tumour progression which is a non-melanoma skin cancer. We first tested the effect of stable transfection of shRNAs to silence the p190RhoGAP expression in A431 human SCC cell line on cell

proliferation in vitro and found that indeed, the clones with the highest silencing had a significantly decreased proliferation rate (Fig. 5a). We then assessed the impact of p190RhoGAP silencing together with the inactivation of p110δ in macrophages on the progression of SCC in vivo. We found that also in the case of SCC, the opposite targeting of p110δ PI3K in macrophages and RhoA in tumour cells abrogated A431 SCC tumour growth

**Fig. 3 The opposite targeting of p110δ and RhoA strongly affects the proliferation, apoptosis and metastasis of melanoma tumour cells. a** Cell proliferation in tumours excised from mice that were treated concomitantly (starting on day +10) with intravenous injections of WT macrophages and intratumoural injections of p190RhoGAP siRNA or intravenous injections of δ$^{D910A/D910A}$ macrophages and intratumoural injections of p190RhoGAP siRNA was determined by BrdU incorporation (brown spots) (right panels). Scale bar = 50 μm. Comparison of BrdU-positive cells in tumours from mice that were treated with intravenous injections of WT macrophages and intratumoural injections of p190RhoGAP siRNA and mice treated with intravenous injections of δ$^{D910A/D910A}$ macrophages and intratumoural injections of p190RhoGAP siRNA (left panel). The symbols on the different groups denote data from eight fields/section of three sections of stained cells. **b** Apoptosis in tumours excised from mice that were treated concomitantly (starting on day +10) with intravenous injections of WT macrophages and intratumoural injections p190RhoGAP siRNA or intravenous injections of δ$^{D910A/D910A}$ macrophages and intratumoural injections of p190RhoGAP siRNA was determined by TUNEL assay (brown spots) (right panels). Scale bar = 50 μm. Comparison of TUNEL positive cells in tumours from mice that were treated with intravenous injections of WT macrophages and intratumoural injections of p190RhoGAP siRNA and mice treated with intravenous injections of δ$^{D910A/D910A}$ macrophages and intratumoural injections of p190RhoGAP siRNA (left panel). The symbols on the different groups denote data from eight fields/ section of three sections of stained cells. **c** Intravasation efficiency of cancer cells as determined by tumour cells blood burden at the end point of the experiment in NSG mice which received WT or δ$^{D910A/D910A}$ macrophages concomitantly (starting on day +10) with intratumoural injections of p190RhoGAP siRNA. Each symbol on the different groups denotes data from a different animal of the respective group. **d** Invasion of cancer cells as determined by immunohistological staining of vimentin (brown) in the lungs of NSG mice which received WT or δ$^{D910A/D910A}$ macrophages concomitantly (starting on day +10) with intratumoural injections of p190RhoGAP siRNA. Scale bar = 50 μm. All immunostainings were performed on tumour sections from tumours excised at the end point of each experiment. All graphs represent means ± s.e.m. Statistically significant differences are indicated by ** ($P < 0.01$) or *** ($P < 0.001$), as determined by the Mann-Whitney $U$ test. **e** Graphical representation of experimental events chronologically.

(Fig. 5b) and metastasis as it was suggested by the abolishment of the tumour cell blood burden (Fig. 5c).

These results provide further evidence for a potentially efficient treatment approach not only in melanoma but also in most common non-melanoma skin cancers such as SCC.

**The efficacy of the combined opposite targeting of p110δ and RhoA correlates with reduced ATX expression.** To assess the potential role of ATX in the positive outcome of the combined targeting of p190RhoGAP and p110δ PI3K in skin cancers we first tested the expression of ATX by IHC. We found that on sections of SCC tumours harvested from mice that received p190RhoGAP siRNA and δ$^{D910A/D910A}$ macrophages both, the abundance of macrophages and the ATX-positive cells in tumour sites were significantly reduced compared with those of mice receiving p190RhoGAP siRNA and WT macrophages (Fig. 6). The results were similar also in case of 451Lu melanoma tumours (Supplementary Fig. 6). This data indicates that either both treatments, the silencing of p190RhoGAP in cancer cells and the inactivation of p110δ PI3K in macrophages or only the one arm of this combined treatment affects the expression of ATX in tumours. Our results showing that the inactivation of p110δ PI3K in macrophages prevents the accumulation of macrophages into tumour sites (Figs. 1b, 6a, b and Supplementary Fig. 6a, b) and that the expression of ATX in tumours of control mice was decreased compared with ATX expressed in tumours from mice receiving p190RhoGAP siRNA and WT macrophages but was increased compared with that in tumours from mice receiving p190RhoGAP siRNA and δ$^{D910A/D910A}$ macrophages (Fig. 6c) together with the fact that macrophages are known important sources of ATX in tumour stroma[102] confirm that the inactivation of p110δ in macrophages which leads to reduced positioning of macrophages to tumour sites contributes to regressed ATX expression (Fig. 6 and Supplementary Fig. 6). Therefore, we then focused to explore whether the increased RhoA activity because of the p190RhoGAP silencing in cancer cells contributes to decreased ATX expression.

Indeed, a siRNA dose dependent decrease of p190RhoGAP expression in SCC A431 cells (Supplementary Figs. 7a and 14a) resulted in a siRNA dose-dependent increase of RhoA-GTP levels (Supplementary Figs. 7b and 14b) and a similar decrease of ATX expression levels (Supplementary Fig. 7c and 14c), indicating that the increased RhoA activity into tumour cells contributes to reduced ATX expression detected on tumour sections (Fig. 6 and

Supplementary Fig. 6). It is of note that the activity of ATX in culture medium of p190RhoGAP siRNA transfected cells was reduced compared to that of mock transfected cells (Supplementary Fig. 7d) and moreover the activity of ATX in plasma from mice receiving p190RhoGAP siRNA and δ$^{D910A/D910A}$ macrophages was found to be decreased compared to that from mice receiving p190RhoGAP siRNA and WT macrophages (Supplementary Fig. 7e) indicating that the secretion of ATX is also affected under both treatment conditions. The reverse experiment showing that pharmacological inhibition of ATX induces the expression levels of RhoA (Fig. 7a, left panel and Supplementary Fig. 9a, left panel) as well as the levels of RhoA-GTP (Fig. 7a, middle panel and Supplementary Fig. 9a, middle panel) confirms an opposite link between RhoA and ATX. Furthermore, the pharmacological inhibition of ATX reduced the phosphorylation levels of Akt (Fig. 7a, right panel and Supplementary Fig. 9a, right panel). Interestingly, the pharmacological inhibition of ATX, although did not affect the expression levels of PTEN (Fig. 7b, right panel and Supplementary Fig. 9b, right panel), induced the activity of PTEN (Fig. 7b, left panel and Supplementary Fig. 9b, right panel) which is most likely a result of the increased RhoA-GTP levels (Fig. 7a, middle panel), and might account for the reduced phosphorylation of Akt (Fig. 7a, right panel). The effect of ATX inactivation on Akt phosphorylation was overcome by pharmacological inhibition of ROCK (Fig. 7c, left panel and Supplementary Fig. 10, left panel), an important effector of RhoA-GTP[103], as well as by a selective small molecule inhibitor of PTEN[104] (Fig. 7c, right panel and Supplementary Fig. 10, right panel) confirming a reverse relationship between ATX and RhoA consisting of a RhoA⊣ATX and an ATX⊣RhoA→PTEN→phospho-Akt link.

**Oral treatment with ATX inhibitor induces ATX expression and fails to prevent tumour growth.** We next explored the potential positive impact of oral administration of the ATX inhibitor PF-8380 (4-[3-(2, 3-dihydro-2-oxo-6-benzoxazolyl)-3-oxopropyl]-(3, 5-dichlorophenyl)methyl ester-1-piperazinecarboxylic acid) or vehicle on tumour growth. PF-8380 is a small molecule specific inhibitor of ATX[105] with subnanomolar potency, good oral availability[105] and no toxic effects on mice[106,107] whereas oral gavages of 30 mg/kg PF-8380 were found to cause a reduction in LPA levels in plasma and inflammatory tissue sites more than 95%[105]. Surprisingly, although the SCC tumour growth was initially prevented by oral administration of PF-8380 then progressively

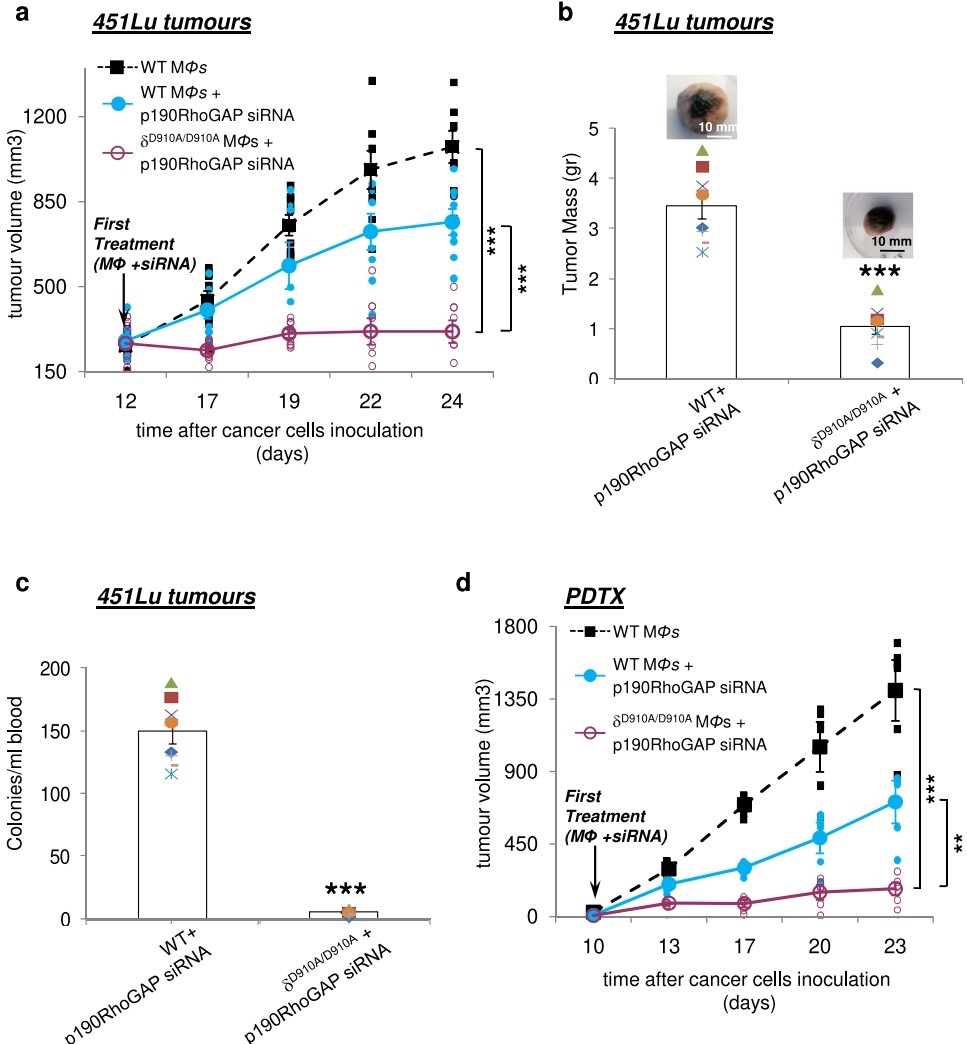

**Fig. 4 BRAF mutations do not affect the efficiency of the combined opposite targeting of p110δ and RhoA in melanoma tumour growth. a** NSG mice were inoculated with 451Lu cells on day 0 and treated with intravenous injections of WT macrophages or WT macrophages and intratumoural injections of p190RhoGAP siRNA or intravenous injections of δ^D910A/D910A macrophages and intratumoural injections of p190RhoGAP siRNA on day +12 and on every other day until the end of the experiment. Tumour growth was measured. *n* = 8 mice/group. **b** Comparison of tumour mass of tumours excised from mice that were treated with intravenous injections of WT macrophages and intratumoural injections of p190RhoGAP siRNA and mice treated with intravenous injections of δ^D910A/D910A macrophages and intratumoural injections of p190RhoGAP siRNA. **c** Intravasation efficiency of 451Lu cancer cells as determined by tumour cells blood burden at the end point of the experiments in NSG mice which received WT or δ^D910A/D910A macrophages concomitantly (starting on day +12) with intratumoural injections of p190RhoGAP siRNA. **d** PDTXs were categorised into the three indicated treatment groups (*n* = 3 mice/group) and were treated during the fourth passage of tumours. The results of two independent treatment groups were combined. Each symbol on the different groups denotes data from a different animal of the respective group. All graphs represent means ± s.e.m. Statistically significant differences are indicated by *** (*P* < 0.001), as determined by the one way ANOVA and the Mann-Whitney *U* test.

the tumour burden reached that of untreated mice (Fig. 8a). Notably, the expression levels of ATX were found to progressively increase in cancer cells from harvested tumours of ATX inhibitor-treated mice (Fig. 8b and Supplementary Fig. 11) whereas in untreated mice the expression of ATX was initially increased but early during tumour progression remained at constantly low levels (Fig. 8b and Supplementary Fig. 11). These results indicate that long term pharmacological inactivation of ATX induces its expression which overcomes the early positive outcome.

## Discussion
Our work shows that a treatment approach consisting of suppressed p190RhoGAP expression into tumours, which keeps RhoA active, combined with inactivation of p110δ PI3K in macrophages almost totally blocks melanoma and SCC progression. It is noteworthy that the efficacy of this combined treatment is independent on BRAF mutational status of melanoma cells since it prevented tumour progression with the same efficiency in melanomas lacking activating mutations in *BRAF* as well as in melanoma tumours harbouring the BRAF(V600E) which is the most frequent mutation in human melanoma[101].

There was an emerging rationale for identifying and targeting new MAPK-independent pathways since the BRAF-signalling targeted therapies were approved to be of limited duration because of the development of drug-resistance disease[108–110]. However, although the targeting of both BRAF and MEK has been used as a standard treatment approach for patients with melanoma harbouring BRAF V600 mutations the challenge still remains because of the variable patient responses to that

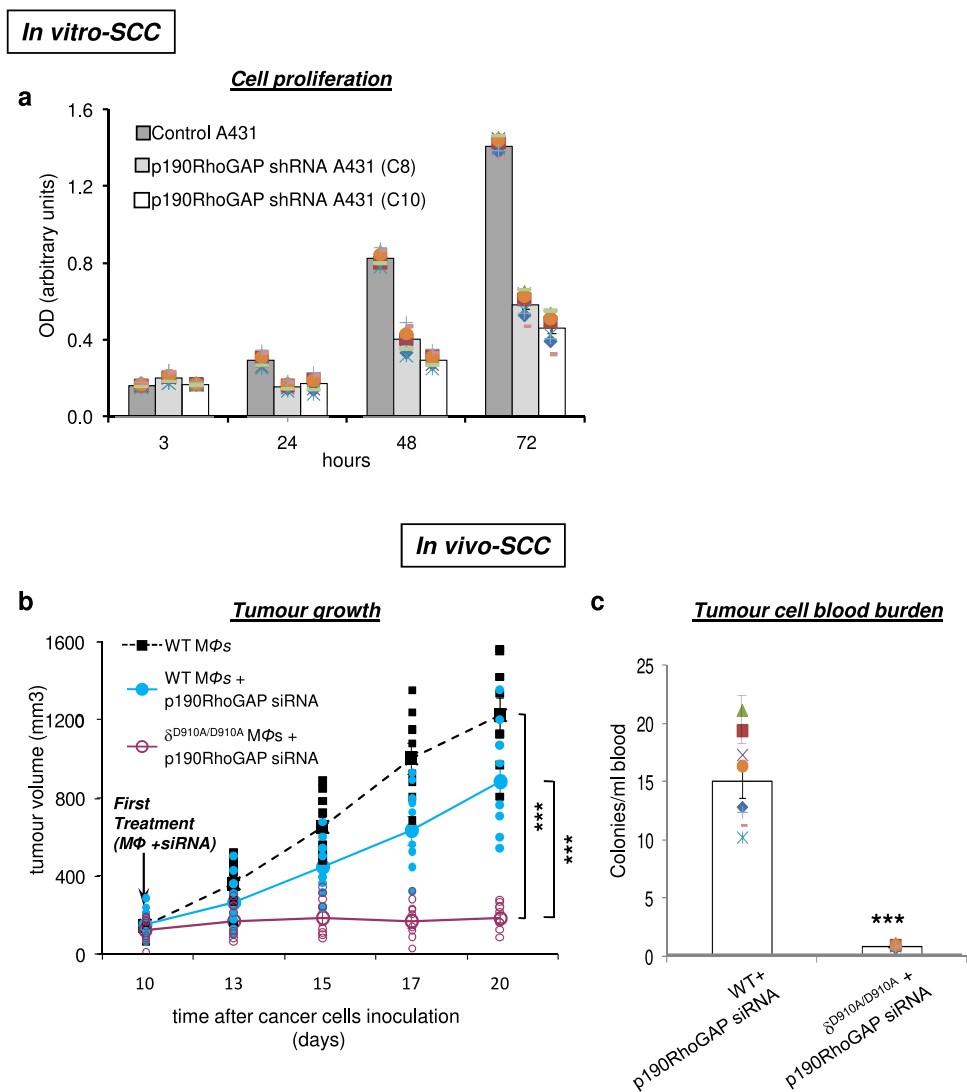

**Fig. 5 A431 SCC tumour growth and metastasis are abolished by the opposite targeting of p110δ and RhoA. a** A431 cells were stably transfected with different p190RhoGAP shRNAs. The proliferation rate in clones (C8 and C10) with the highest silencing efficacy was determined and compared with that of control A431 cells. The symbols on the different groups denote data from three different experiments performed in triplicate. **b** NSG mice were inoculated with A431 cells on day 0 and treated with intravenous injections of WT macrophages or intravenous injections of WT macrophages and intratumoural injections of p190RhoGAP siRNA or intravenous injections of δ$^{D910A/D910A}$ macrophages and intratumoural injections of p190RhoGAP siRNA on day +10 and on every other day until the end of the experiment. Tumour growth was measured. $n = 9$ mice/group. Each symbol on the different groups denotes data from a different animal of the respective group. **c** Intravasation efficiency of A431 cells as determined by tumour cells blood burden at the end point of the experiments in NSG mice ($n = 9$ mice/group) which received WT or δ$^{D910A/D910A}$ macrophages (starting on day +10) and intratumoural injections of p190RhoGAP siRNA. Each symbol on the different groups denotes data from a different animal of the respective group. All graphs represent means ± s.e.m. Statistically significant differences are indicated by *** ($P < 0.001$), as determined by the one-way ANOVA and the Mann-Whitney $U$ test.

combination and of drug resistance[111–113]. On the other hand, the attack of the cross-talk between the MAPK and PI3K/AKT pathways by targeting both pathways constitute a promising approach though the development of dose-limiting toxicities[114] makes it also challenging and emerges the need for new combinations. Moreover, the activation of PI3K pathway has been considered as a mediator of resistance to BRAF(V600E)-pathway targeted therapies[45,115]. Besides, the critical role of PI3K in the interplay between cells of TME, including TAMs, and cancer cells[83,116] has also become a promising research hotspot in different cancers. Indeed, PI3K has been implicated in the polarisation and reprogramming of TAMs[117,118], in the functional communication of TAMs with cancer cells and in drug resistance, among others, in colorectal cancer, colitis-induced cancer, ovarian and breast cancer[119–122]. Especially for breast cancer, we have

recently documented that the inactivation of p110δ in macrophages is sufficient to prevent the localisation of macrophages into tumour stroma and consequently to suppress tumour growth and metastasis[73]. Therefore, the efficacy of the combined opposite targeting of p110δ PI3K and RhoA in preventing melanoma tumour growth and metastasis strongly provides a major advance for designing new successful therapies and for considering the use of p110δ-selective inhibitors as combinatorial regimens in melanoma treatment.

Skin cancer cells and other cancer cells are known to produce high amounts of ATX[123–125], a secreted enzyme which converts extracellular lysophosphatidylcholine (LPC) into LPA[18,19,126] which then activates at least six G protein-coupled receptors to increase cell division, survival and migration[20,21,127–132]. Therefore, ATX has been shown to play an important role in

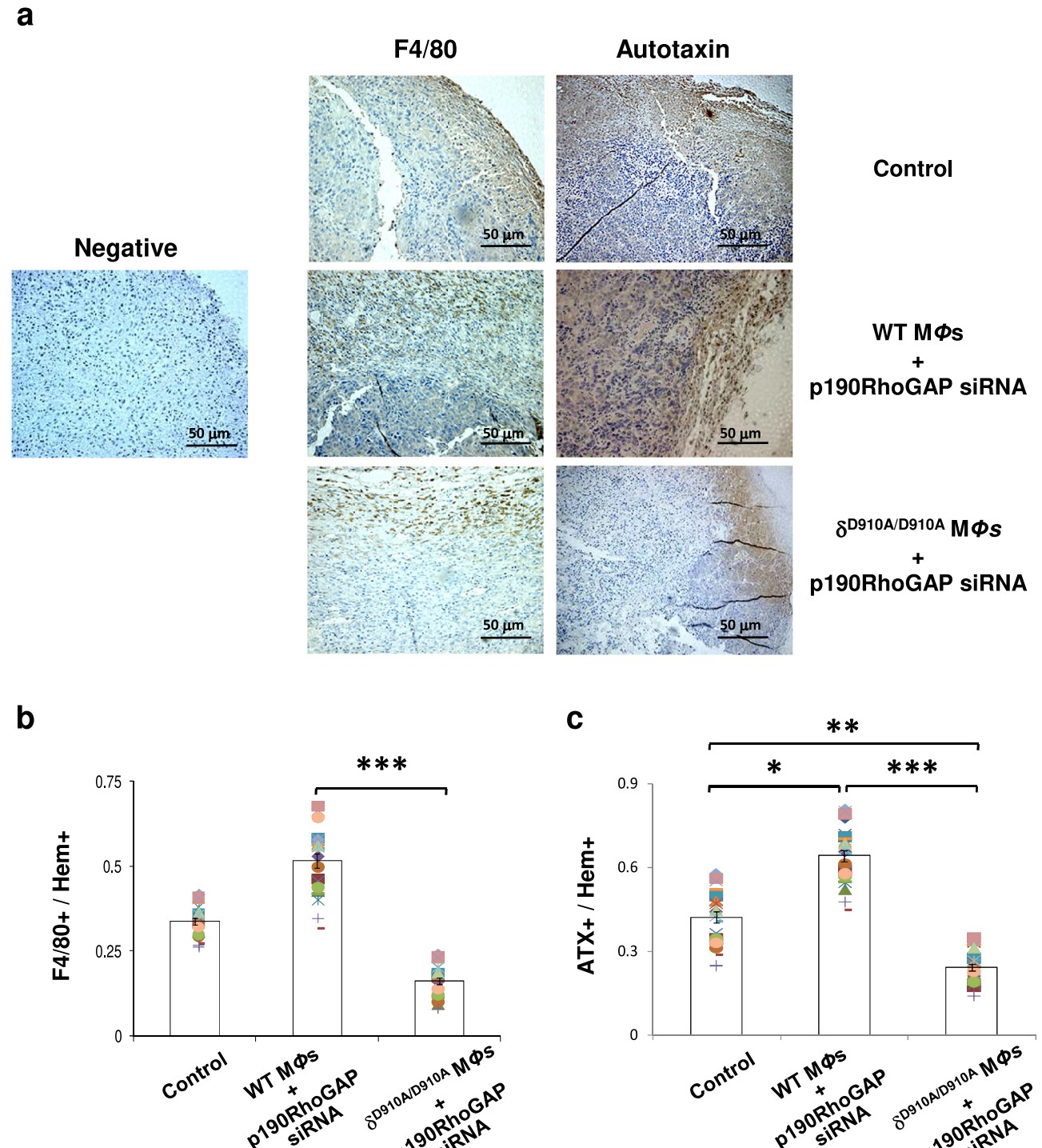

**Fig. 6 Impact of the opposite targeting of p110δ and RhoA on the recruitment of macrophages to tumour sites and on ATX expression in SCC tumours.**
**a** Representative images of immunohistochemical staining with anti-F4/80 antibody (left panels) or anti-ATX antibody (right panels) and Hematoxylin (blue) in representative sections of SCC tumours from NSG mice received no treatment (Control) or intravenous injections of WT macrophages and intratumoural injections of p190RhoGAP siRNA or intravenous injections of δ$^{D910A/D910A}$ macrophages and intratumoural injections of p190RhoGAP siRNA. $n = 8$ mice/group. Scale bar = 50 μm. **b** Comparison of F4/80-positive cells in tumours from control mice or mice received intravenous injections of WT macrophages and intratumoural injections of p190RhoGAP siRNA or intravenous injections of δ$^{D910A/D910A}$ macrophages and intratumoural injections of p190RhoGAP siRNA. **c** Comparison of ATX-positive cells in tumours from control mice or mice received intravenous injections of WT macrophages and intratumoural injections of p190RhoGAP siRNA or intravenous injections of δ$^{D910A/D910A}$ macrophages and intratumoural injections of p190RhoGAP siRNA. All immunostainings were performed on tumour sections from tumours excised at the end point of each experiment. The symbols on the different groups denote data from eight fields/section of three sections of stained cells. All graphs represent the mean ± s.e.m. of three separate experiments. Statistically significant differences are indicated by * ($P < 0.5$) or ** ($P < 0.01$) or *** ($P < 0.001$), as determined by the ANOVA and Mann–Whitney $U$ test.

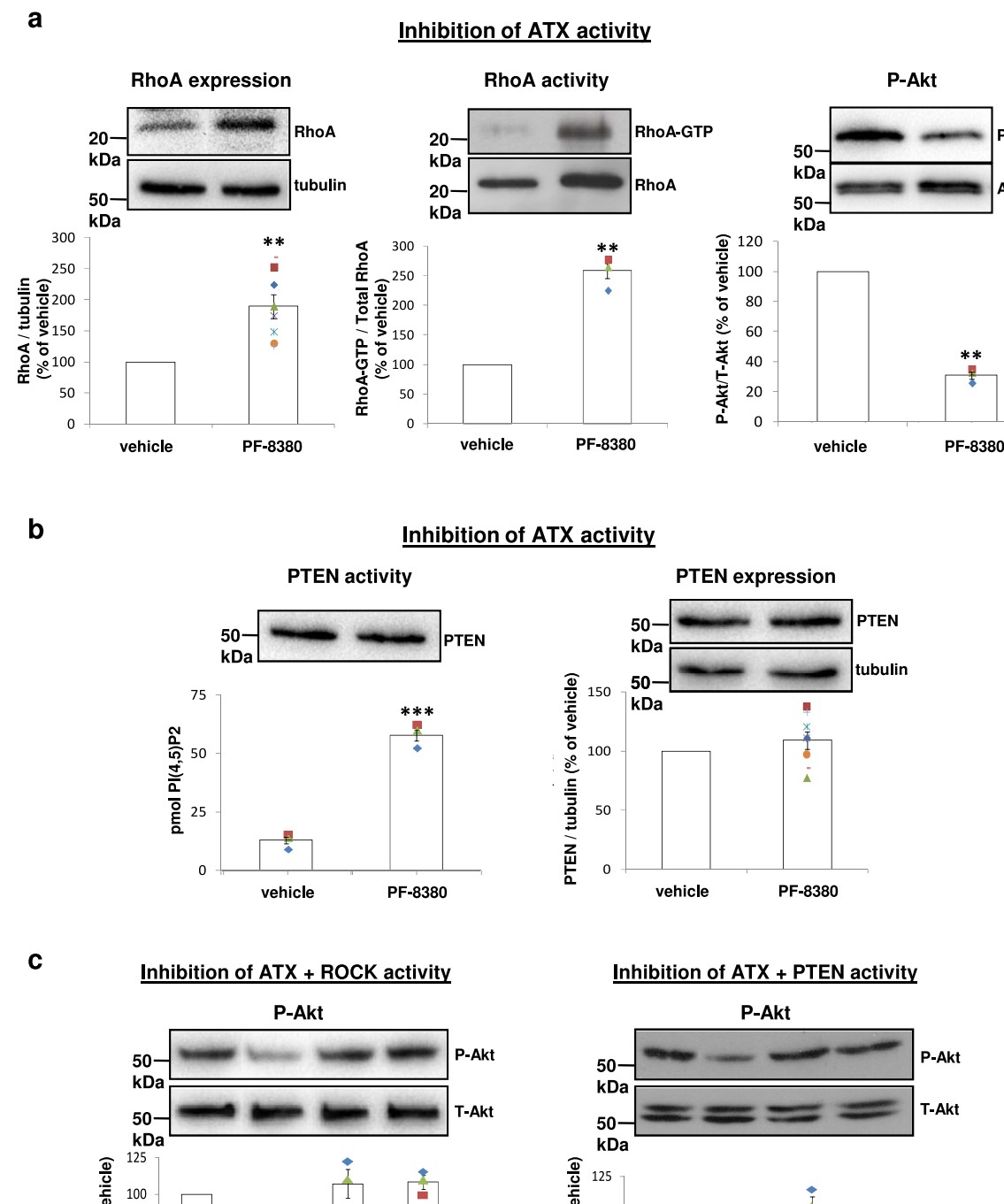

inflammation, tumour growth, metastasis and chemo-resistance[22–26]. Furthermore, LPA exerts direct effects not only on tumour cells but also on the tumour microenvironment and on the other hand, ATX is also produced by tumour-associated stroma even in cancers that produce little ATX[35,126,133]. Thus, increased ATX enzymatic activity increases tumour LPA concentrations, sustaining a cycle that fuels tumour growth and

metastasis. TAMs are considered to play an essential role as important producers of ATX and LPA directly or indirectly by producing inflammatory mediators that stimulate tumour stroma cells to increase ATX production[36,102,134]. Despite the negative role of ATX in tumour growth, ATX signalling protects cancers cells from cytotoxic effects of radiotherapy and chemotherapy whereas it has been additionally shown that inhibition of ATX

**Fig. 7 Inhibition of ATX activity affects RhoA, pAkt and PTEN activity. a** Inhibition of ATX activity by 1 μM PF-8380 induces the expression and activity of RhoA whereas reduces the phosphorylated Akt levels in SCC cells. The bands of total RhoA presented for normalisation in the middle panel were derived from different membrane but from the same cell lysates whereas in left and right panels the bands of tubulin and total Akt, respectively were derived from the same respective membranes. **b** The activity of PTEN is induced whereas the expression of PTEN remains unaffected by the inhibition of ATX activity. The tubulin bands presented for normalisation were derived from the same membranes. **c** Inhibition of ROCK activity by 25 μM of the ROCK inhibitor Y27632 or inhibition of PTEN activity by 500 nM of the PTEN inhibitor VO-OHpic overcomes the biological effects on pAkt of ATX inactivation. The total Akt bands presented for normalisation were derived from the same respective membranes. The symbols on the different groups denote data from at least three different experiments. All graphs represent the mean ± s.e.m. of at least three separate experiments. Statistically significant differences are indicated by ** (P < 0.01) or *** (P < 0.001), as determined by the ANOVA and Mann-Whitney U test.

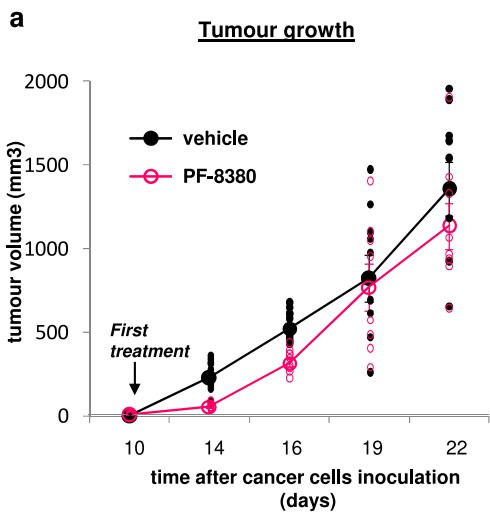

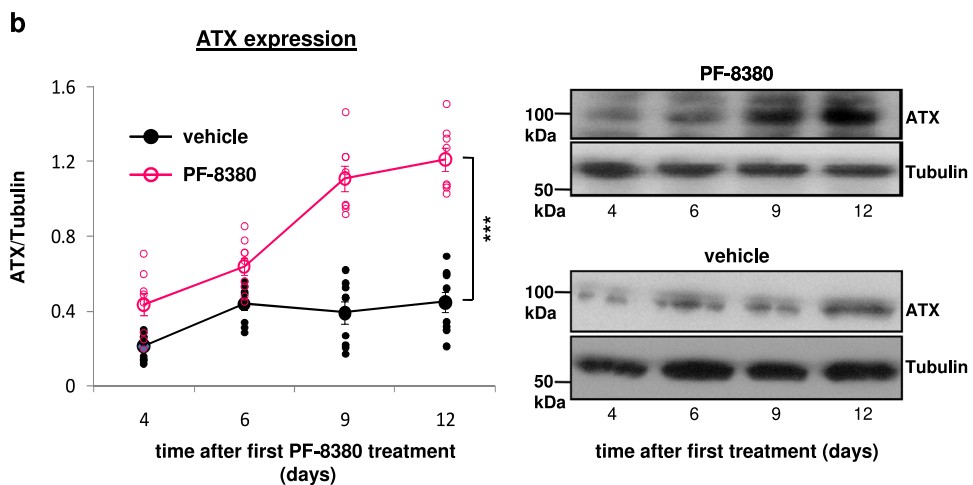

**Fig. 8 Pharmacological inhibition of ATX temporarily prevents tumour growth whereas progressively induces the expression of ATX. a** SCC tumour bearing mice were treated twice daily *per os* with vehicle or PF-8380 (30 mg/kg) from day +10. n = 8–9 mice/group. **b** Cancer cells harvested from tumours of vehicle or PF-8380-treated SCC tumour-bearing mice (n = 8–9 mice/group) were examined for their expression levels of ATX. The tubulin bands presented for normalisation were derived from the same respective membranes. Each symbol on the different groups denotes data from a different animal of the respective group. Graphs represent the mean ± s.e.m. Statistically significant differences are indicated by *** (P < 0.001) as determined by the ANOVA and Mann-Whitney U test.

activity increases ATX expression through feedback regulation[126,135]. Therefore, it seems that a delicate balance in ATX inhibition activity is needed for an efficient outcome.

Our present results clearly show that the effectiveness of the combined opposite targeting of p110δ PI3K and RhoA in preventing skin tumour growth is a results of its efficacy to suppress the expression of ATX, which derives from both, the modulation of macrophage recruitment to tumour sites, because of p110δ

PI3K inactivation, and the silencing of p190RhoGAP expression in tumour cells. TAMs are known to play an essential role as main sources of ATX in tumour stroma[36,102,134]. At present, it is not clear how p190RhoGAP has a positive or RhoA has a negative impact on ATX expression since ATX is regulated by complicated mechanisms at transcriptional and post-transcriptional levels[136]. However, given that increased RhoA activity resulted in increased PTEN activity and consequently in reduced phosphorylation of

Akt, a molecular link that could explain the negative impact of RhoA activity on ATX expression is the Akt-regulated ATX export from the endoplasmic reticulum[136,137]. The reduced ATX expression upon increased RhoA activity into tumours and on the other hand the induced RhoA expression and activity upon inhibition of ATX activity in cancer cells emerges a negative reverse link between RhoA and ATX. It is of note, that pharmacological inhibition of ATX activity *per os* triggered a positive feedback regulation that induced the expression of ATX in tumour cells and consequently failed to prevent tumour growth while the combined targeting of p190RhoGAP and p110δ PI3K blocked tumour growth by suppressing the expression of ATX without triggering this feedback mechanism.

Overall, we provide evidence that a combined treatment approach consisting of suppressed p190RhoGAP expression into tumours and inactivation of p110δ PI3K in macrophages blocks melanoma and SCC progression by decreasing ATX expression levels. These results have several potential implications. First, ATX seems to have a key role as a target molecule to combat melanoma and SCC progression however, its importance diminishes when direct pharmacological inhibition is used. Furthermore, the expression levels of p190RhoGAP are emerged as a prognostic factor in patients with skin cancer and moreover, promising therapeutic strategies are paved for potential targeted therapies against p190RhoGAP which indirectly will suppress ATX in cancer cells. Finally, this work further illustrates that the activity of p110δ PI3K in macrophages is a key contributor to skin cancer progression pointing the use of p110δ selective inhibitors as a combinatorial drug regimen for skin cancer therapy. Collectively, our results reveal precise and potentially effective cellular and molecular targets for cancer therapy.

## Methods

**Chemicals**. The p190RhoGAP siRNA was purchased from Origene (Arhgap35, Mouse-3 unique 27mer siRNA duplexes, # SR422929) or from Dharmacon (ON-TARGETplus SMARTpool Mouse Arhgap35 siRNA, # L-055189). The GRLF1 (ARHGAP35) Human shRNA Plasmid Kit was from Origene (#TR312603). The PI3K p110δ inhibitor IC87114 was purchased from Chemietek and re-suspended in 30% PEG-400, 0.5% Tween-80 and 5% propylene glycol for experiments in mice.

**Antibodies**. Antibodies to pAkt (#4060) and Akt (#4691) were from Cell Signaling Technology Inc., and were used at a concentration of 1/1000. The monoclonal mouse antibodies to RhoA (#sc-418) and PTEN (#A2B1) and polyclonal anti-PTEN (#sc-6818) were from Santa Cruz Biotechnology (Santa Cruz, CA) and were used at a concentration of 1/100. Monoclonal anti-Rac1 (#05-389) was from Millipore and was used at 1/500. Other sources for antibodies were as follows: F4/80 (Bio-Rad #MCA497GA), ATX (rat, 4F1 clone, D323-3, MBL), vimentin (Thermoscientific #RM-9120). The concentrations of those antibodies used are 1/50 for F4/80, 1:40 for ATX and 1/200 for vimentin.

**Cell culture**. The B16.F10 murine metastatic melanoma and A431 human squamous cell carcinoma (SCC) cell lines were a gift from Prof. Papamatheakis (Institute of Molecular Biology and Biotechnology, Crete, Greece). The 451Lu human metastatic melanoma cell line was kindly provided by Prof. Marcel Deckert (Université Côte d'Azur, Centre Méditerranéen de Médecine Moléculaire). The cell culture medium consisted of DMEM (Invitrogen, Life Technologies), 10% FCS and 1% penicillin–streptomycin. Cells were cultured at 37 °C with 5% $CO_2$ and regularly tested for mycoplasma infection. Cells were

cultured to 70% confluence, split, counted and resuspended in sterile PBS prior to injection into mice. For in vitro experiments, the inhibitors of ATX (PF-8380), ROCK (Y27632) and PTEN (VO-OHpic) activity were used at concentrations of 1 μM, 25 μM and 500 nM, respectively[91,138].

**Mice**. Since the incidence of skin cancer and mortality rate is higher in men than in women worldwide[1,139,140], all experiments were carried out using male mice. All mice were housed in a pathogen-free animal facility at the Animal Facilities of the Medical School, University of Crete in Heraklion, Crete. All experimental procedures on mice were first approved by the Research Animal Care Committee of Medical School, University of Crete and by the Veterinary Department of Heraklion Prefecture (Protocol No 54827). Male NOD.$Cg$-$Prkdc^{scid}$ $Il2rg^{tm1Wjl}$ (NOD $scid$ gamma or NSG) mice were obtained from Taconic Biosciences A/S and from the Institute of Molecular Biology and Biotechnology, Crete, Greece. BALB/c nude male mice were obtained from Charles River Laboratories. δ$^{D910A/D910A}$ mice were kindly provided by Prof. Bart Vanhaesebroeck (UCL).

Mice used in all experiments were male, age-matched (2–3 months of age) and in excellent physical health. The selection of the appropriate group sample size was based on calculations using the NC3Rs recommended Resource Equation method, pilot experiments and prior knowledge of the variability of skin tumours in NSG and BALB/c nude mice. The animals were distributed to experimental groups by simple random sampling at the beginning of each experiment. All mice received injections of tumour cells and then equal number of mice was selected at random for each treatment arm. Each experiment was performed using a minimum of 6 mice per group (the exact number of mice is indicated in each figure legend) and was repeated at least three times with independent groups of animals to assess reproducibility.

**In vivo studies of tumour growth in mice**. Anaesthetised NSG mice were inoculated subcutaneously on day 0 with $10^6$ cells in 100 μl PBS. Mice were then randomly assigned into the control group receiving vehicle alone or into the treatment groups receiving intratumoural injections of the p190RhoGAP siRNA and/or intravenous injections of macrophages as described below. When this is indicated, the PI3K p110δ inhibitor IC87114 (35 mg/kg) was administered in B16 tumour-bearing Balb/c nude mice by oral gavages once daily[73]. The ATX inhibitor PF-8380 (30 mg/kg) was administered in SCC tumour bearing mice by oral gavages twice daily[106,107]. Tumour growth was monitored by measurement of the longest perpendicular tumour diameters using a digital calliper every 3–6 days. The tumour volume ($V$) was calculated[141] as $V$ (mm$^3$) = length (mm) × width (mm)$^2$ × 0.5. At the end of the study, animals were euthanized and primary tumours and lungs were removed for the experiments described below.

**Mice-bearing melanoma patient-derived xenograft (PDX)**. PDTXs were developed by the implantation of human melanoma specimen, following surgical removal from patient's tumour into an NSG mouse[142]. Patients gave their written informed consent and the procedures were approved by the Board of Directors of General University Hospital of Crete (Decision No 987). Briefly, the viable tumour was dissected into small pieces of about 3–8 mm$^3$ prior to implantation and directly transplanted subcutaneously into a mouse which was designated generation 0. The time for palpable tumour to develop was 2–3 months. Upon engraftment of tumours in the first cohort of recipient mice, the growing tumours were removed and grafted onto another cohort (usually from 1 mouse to 3 mice) that was designated generation

1 and then serially over several passages. Experiments were conducted on the second or third generation[142]. PDTX mice were randomly divided into treatment groups ($n = 3$ mice per group). The treatment conditions and tumour burden monitoring were performed as described above.

**Induction of RhoA activity into tumours**. RhoA was kept active into tumours by suppression of p190RhoGAP expression using intratumoural p190RhoGAP siRNA injections. p190RhoGAP is a potent inhibitor of RhoA GTPase by catalysing the return of RhoA-GTP (active state) to RhoA-GDP (inactive state)[89]. p190RhoGAP siRNA (0.8 mg/kg) was complexed with oligofectamine[143] and administered by intratumoural injections starting on day +10 when the tumour was palpable.

**Adoptive macrophage transfer**. For the adoptive macrophage transfer experiments, wild-type (WT) mice or $\delta^{D910A/D910A}$ mice in which p110δ alleles are replaced by a kinase-dead version of p110δ mutated in the ATP binding site[92], were served as macrophage donor mice. WT and $\delta^{D910A/D910A}$ mice received intraperitoneal injections of 5 ml of 3 % Brewer thioglycollate medium and 1 week later were euthanized and peritoneal macrophages were collected[144]. On day 10 or when this is indicated, on day +2 or on day +18, $5 \times 10^5$ macrophages from WT or $\delta^{D910A/D910A}$ mice were intravenously administered into tumour-bearing NSG mice. The adoptive macrophage transfer was repeated every 48 h[73]. Tumour growth was monitored and calculated as described above. At the end of the study, mice were anaesthetised and tumour cell blood burden was determined as described below, followed by sacrifice of the mice and removal of the primary tumour and the lungs for the experiments described below.

**Tumour cell blood burden**. Mice were anaesthetised and blood was collected from the right atrium by a syringe coated with heparin via heart puncture with a 25-gauge needle. The blood was plated into culture dishes filled with 5% foetal bovine serum/αMEM growth medium which the next day was replaced with fresh medium containing geneticin (0.8 mg/mL) to selectively grow the tumour cells[97]. After 3–7 days tumour cell clones in the dish were counted and the tumour blood burden was calculated as the number of colonies in the dish divided by the volume of the blood taken[97].

**Measurement of ATX activity**. The blood was collected as described above and the activity of ATX was measured in ten µl of plasma[145]. Briefly, 15 µl of buffer A consisting of 100 mM Tris-HCl, pH 9.0; 500 mM NaCl; 500 mM MgCl$_2$; and 0.05% v/v Triton X-100 was mixed with 10 µl of plasma, incubated for 30 min at 37 °C and then 25 µl of a solution containing 6 mM C14:0-LPC in buffer A was added into the samples and further incubated for 6 h at 37 °C. 20 µl of each sample was placed into a 96-well plate followed by the addition of 90 µl/well of buffer C consisting of 9.65 ml buffer B (100 mM Tris-HCl, pH 8.5, and 5 mM CaCl$_2$), 110 µl of 30 mM TOOS (N-ethyl-N-(2-hydroxy-3-sulfopropyl)-3- methylaniline, 110 µl of 50 mM 4-aminotipyrine, 6.6 µl of 1000 U/ml horseradish peroxidase, and 110 µl of 300 U/ml choline oxidase and were incubated for 20 min at 37 °C. The formation of choline was measured at 550 nm. The activity of ATX represents the choline released from LPC divided by the volume of the plasma taken[145]. Similar procedure was followed when the activity of ATX was determined in cell culture medium.

**Immunohistochemistry**. The deparaffinization and dehydration of tissue sections were carried out through graded ethanol series.

To inactivate endogenous peroxidase activity the sections were treated with 3% hydrogen peroxide for 30 min at 20 °C and rinsed in PBS. Epitope unmasking was performed by heating for 40 min in citrate buffer (pH6.0) (in the case of anti-vimentin and anti-ATX). Blocking for non specific binding was performed using 1% BSA for 1 h at room temperature, followed by incubation with the primary antibodies overnight at 4 °C (1/50 for F4/80 (Bio-Rad #MCA497GA), 1:40 for ATX (rat, 4F1 clone, D323-3, MBL) and 1/200 for vimentin (Thermoscientific #RM-9120)). The tissue sections were then incubated with an anti-rat (for F4/80 and ATX) or an anti-rabbit HRP (for vimentin) with DAB and H$_2$O$_2$, counterstained with Hematoxylin and mounted with Vectashield mounting medium (Vector Labs)[73,146].

The counting of ATX-positive and F4/80-positive cells was performed using ImageJ software (NIH). Vimentin was quantified as the average pixel intensity per field of view using Scion Image freeware (Scion Corp., Frederick, MD, USA)[147]. Values are presented as means ± s.e.m of stained cells counted from 5–8 fields/section (randomly selected) and from three sections/determination. The immunohistochemisty experiments were performed using tissues from at least three repeat experiments. The results were similar among different experiments.

**BrdU incorporation**. To determine the proliferative rate of tumour cells, a BrdU staining kit (Millipore #2760) was used according to the manufacturer's instructions. Briefly, mice were injected intraperitoneally with 100 mg/kg bodyweight of 5-bromo-2′-deoxyuridine (BrdU; Calbiochem) 2 h before euthanasia and then the BrdU-positive tumour cells were detected. The counting of BrdU-positive cells was performed using ImageJ software (NIH). Values are presented as means ± s.e.m. of BrdU-positive/Hematoxylin stained cells counted from 5 to 8 fields/section (randomly selected) and three sections/measurement. The BrdU incorporation procedure was performed on tissues obtained from at least three separate experimental groups of animals for each treatment condition. The results were similar among the different experimental repeats.

**TUNEL assay**. The DeadEnd colorimetric TUNEL system (Promega #G7130) was used for the detection of apoptosis in tumour cells following the procedure described in the manufacturer's instructions. The counting of TUNEL-positive cells was performed using ImageJ software (NIH). Values are presented as means ± s.e.m. of TUNEL-positive/Hematoxylin stained cells counted from 5 to 8 fields/section (randomly selected) and three sections/measurement. The TUNEL procedure was performed on tissues obtained from at least three separate experimental groups of animals for each treatment condition. The results were similar among different experiments.

**Determination of GTP-loading on RhoA and Rac1**. The RhoA activation assay was performed using GST-RBD (Rho binding domain of Rhotekin expressed as a GST fusion protein) (Cytoskeleton Inc). Cells were lysed in RIPA buffer (50 mM Tris.HCl pH 7.2, 1 mM EDTA, 1% Triton X-100, 0.5% sodium deoxycholate, 0.1% SDS, 500 mM NaCl, 10 mM MgCl$_2$, 10% glycerol supplemented with protease inhibitors). Cleared cell lysates were incubated at 4 °C for 1 h with 50 µl glutathione-Sepharose-bound GST-RBD. Precipitates were washed three times with washing buffer (50 mM Tris.HCl pH 7.2, 1% Triton X-100, 150 mM NaCl, 10 mM MgCl$_2$, supplemented with protease inhibitors) and suspended in Laemmli sample buffer followed by SDS-PAGE and WB for RhoA using a monoclonal antibody. The Rac1 activation assay was performed using GST-PBD (p21-binding domain of PAK, expressed as a GST-fusion protein). Cells were lysed in

Mg2+ lysis buffer, provided in the assay kit, mixed with 8 μg GST-PBD bound to glutathioneagarose and incubated for 1 h at 4 °C. Precipitates were washed three times with Mg2+ lysis buffer and suspended in Laemmli sample buffer. Proteins were separated by 12% SDS PAGE, transferred to PVDF membrane and blotted with anti-Rac1 antibody.

**Isolation of TAMs and cancer cells from tumours**. The tumours were dissected, the lymph nodes, necrotic tumours, fat and vessels around the tumour were removed, for an optimal tumour-associated macrophage purity, added on a petri dish and digested with dissociation buffer containing RPMI 1640 cell culture medium, 5% FBS, Collagenase/hyaluronidase and DNase I (10 U/mL) and then TAMs were isolated[148]. For the isolation of cancer cells from tumours an enzyme mixture consisting of 0.05 mg/mL Collagenase I, 0.05 mg/mL Collagenase IV and 0.01 mg/mL DNase I in HBSS was used for digestion and dissociation followed by the isolation of cancer cells using the Tumor cell isolation kit (Cell Biolabs).

**Western blotting analysis**. For western blot analysis, a lysis buffer containing 150 mM NaCl, 1.5 mM MgCl$_2$, 1 mM EGTA, 10% glycerol, 100 mM NaF, 25 mM glycerolphosphate, 1% IPE-GAL, 1 mM DTT, 10 mM Na-pyrophosphate, 1 mM PMSF, 10 μg/ml aprotinin, 10 mM Na$_4$VO$_3$ and 50 mM Hepes, pH 7.4 (in the case of the PTEN lipid phosphatase activity assay) or a lysis buffer containing 20 mM Tris-HCl, pH 7.4, 137 mM NaCl, 1 mM CaCl$_2$, 1 mM MgCl$_2$, 1 mM sodium orthovanadate, 1% NP-40 and 1 mM PMSF were used for cell lysis followed by clearing of the lysates by centrifugation in a cooled microcentrifuge. Supernatants were directly immunoprecipitated (for PTEN assay) at 4 °C overnight using an anti-PTEN (Santa Cruz Biotechnology #A2B1 or #sc-6818). Immune complexes were collected with 50% slurry of protein A-Sepharose after incubation for 2–3 h at 4 °C and washed according to the manufacturer's (Echelon Biosciences) instructions provided with the ELISA kit. For analysis of total cell lysates by western blotting, 50–70 μg of cell extract was loaded per lane on an SDS-PAGE gel and transferred onto PVDF membranes. The blots were probed with the indicated antibodies followed by detection using enhanced chemiluminescence (GE Healthcare). The normalisation of each protein signal was performed by using the signal of tubulin or the respective protein that was derived from the same membrane or different membranes but from the same cell lysates.

**PTEN lipid phosphatase activity assay**. The lipid phosphatase activity of PTEN was measured by ELISA, according to the manufacturer's (Echelon Biosciences) instructions. Briefly, PTEN was immunoprecipitated from cell lysates that were derived from tumour cells isolated from each tumour-bearing mouse from each experimental group and the PI(4,5)P2 produced by the immunoprecipitates was determined by comparison to a standard curve consisting of PI(4,5)P2 standards bound to the ELISA plate. To ensure that the appropriate amount of enzyme was used so that the produced PI(4,5)P2 was in the range of the respective standard curve, different amounts of proteins of cell lysates, from which PTEN was immunoprecipitated, were tested in pilot experiments.

**In vitro shRNA and siRNA transfection method**. Briefly, for shRNA transfection, *E. coli* cells were transformed by heat shock with five plasmid constructs (A, B, C, D and NT) containing shRNA for p190RhoGAP in lentiviral GFP plasmid vector (4 + 1 scrambled). The ARHGAP15 human, four unique 29mer shRNA constructs in pGFP-C-shLenti vector were purchased

from Origene (cat no TR312603). The colonies left to growth up and plasmid DNA was purified using a Plasmid DNA purification kit. Virus was produced by transfection of plasmid DNA and pVSV-G and pΔ8.1 (virus capsid) into HEK293T cells, then the virus particles were collected and used for transfection of SCC cells followed by construction of different cancer cell lines (and colonies) of different shRNAs. For p190RhoGap silencing a cocktail of p190RhoGap specific siRNAs and Interferin transfection reagent were used according to the manufacturers' instructions.

**In vitro proliferation assay**. Cell proliferation of p190RhoGAP shRNA transfected SCC cultures was measured using the CellTiter 96® AQueous Assay (Promega) according to manufacturers' instructions.

**Statistics and reproducibility**. Error bars displayed throughout the manuscript represent s.e.m. and were calculated from technical or biological replicates as described in the figure legends and in the respective methods. Data shown are representative of at least three independent experiments, including animal studies, histological images, blots and gels. Data were analysed using the STATISTICA 7 statistical software package. Statistical significance was determined using ANOVA to determine whether there are any statistically significant differences between the means of three or more groups and then we used the non-parametric Mann-Whitney test to compare the two groups of interest.; $*P < 0.05$; $**P < 0.01$; $***P < 0.001$.

### Data availability
The authors declare that the data supporting the findings of this study are available within the paper and its Supplementary Information. Source data underlying figures are provided in Supplementary Data 1.

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

## Acknowledgements

This research was supported by the Hellenic Foundation for Research and Innovation (H.F.R.I.) under the "First Call for H.F.R.I. Research Projects to support Faculty members and Researchers and the procurement of high-cost research equipment grant" (Project Number: 3405) (EAP) and the Programme NSRF 2007–2013, "Education &Lifelong Learning" (Action ARISTEIA II, Project No. 4078) of the Ministry of Education and Religious Affairs, Greece, co-financed by Greece and the European Union (EAP).

## Author contributions

N.T., L.X. and E.G. performed experiments with mice and data analyses and drafted the manuscript. A.T., I.V., A.A. and G.V. performed in vitro experiments. A.B. performed immunohistochemistry. M.T. performed histology and interpreted histopathology and immunohistochemistry. Clinical samples and clinical information were obtained from E.D.B. A.M. supervised experiments. E.A.P. conceived the project, designed and analysed the experiments, supervised experiments and wrote the paper.

## Competing interests

The authors declare no competing interests.
