## [Peer Review File · Communications Biology]

Reviewers' comments:

Reviewer #1 (Remarks to the Author):

In this article, Tzenali et al described a novel combinative intervention for skin cancers (melanoma and squamous cell carcinoma) that targets a dialog between tumour cells and macrophages leading to tumour growth and metastatic dissemination. The data are convincing, the results and material&method well written; the conclusions are well substantiated and correspond to the data shown. In my opinion, this article will benefit to the clinical community by translating basic science findings (from the authors) into a meaningful therapeutic strategy in melanoma patients. However, a few points should be addressed before publishing.

Major points:

- Introduction is too succinct; in particular, it is lacking important background on SCC and the proteins studied PI3K/PTEN, Rho and ATX; what is the meaning of the abbreviation SCC?
- Figure 1 To me, the macrophages are not equally distributed (around the tumours?); provide higher resolution and larger enlargements. Are the quantifications performed on the whole slide? Are there differences in pAKT activation in macrophages in both conditions;
- Provide Rho activity experiments;
- p6 show data on PDTXs models;
- Demonstrate the role of ATX in the loop between tumoural Rho and macrophage's PI3Kdelta;
- Fig 4C + M&M: colony assay: how are cells grown ex vivo – do they maintain the same levels of activation of PTEN, Rho and ATX?
- Discussion: recent publications on PI3K-dependent macrophage / tumour cell dialog in other cancer settings are missing in the discussion and could further argue for the importance of the results of the authors.

Minor:

- Some scales are missing in some photos.

Reviewer #2 (Remarks to the Author):

Tzenaki et al. propose a novel combinatorial regimen for the treatment of skin cancers by targeting p190RhoGAP and p110 δ PI3K in both melanoma and squamous cell carcinoma. The authors show that inactivation of p110 δ PI3K in macrophages blocks melanoma and SCC progression and that this effect may be enhanced by silencing p190RhoGAP expression. Furthermore, it was found that these results in melanoma were independent of BRAF mutation status.

The combinatorial approach by Tzenaki et al. shows a strong effect on tumor progression in vivo in both melanoma and squamous cell carcinoma with some promising results. However, key experiments lack important controls, and some claims are made without sufficient supporting evidence. Please see the following comments:

Major points:

1. Figures 2-6: It is not clear why the authors use p190RhoGAP siRNA without controls in the in vivo studies with the exception of Fig. 2A. It cannot be concluded that targeting p110 δ PI3K and p190RhoGAP has a combined effect on tumor growth if p190RhoGAP expression is silenced in all conditions. Non-targeting siRNA controls should be included if the authors are to conclude that the results are due to a combined effect of δ D910A/D910A macrophages and silencing p190RhoGAP and not just the δ D910A/D910A macrophages.
2. It cannot be claimed that the effects of silencing p190RhoGAP are RhoA-dependent without testing the impact of silencing p190RhoGAP on the activity of other Rho GTPases. While p190RhoGAP is the predominant RhoA GAP, it can also function as a GAP for Rac1 in cells and possibly others (for example, see <https://pubmed.ncbi.nlm.nih.gov/27481945/> and

<https://pubmed.ncbi.nlm.nih.gov/23499677/>). Either additional experiments should be included to rule out that p190RhoGAP is acting on any other Rho GTPases in these cancer cell lines, or it should be noted that other Rho GTPases may be involved. Subsequently, the authors should state that they are targeting p190RhoGAP in the title and in the text, not RhoA, since RhoA exclusivity has not been conclusively shown.

3. Figure 7A, middle panel and 7C, right panel: The effects on P-Akt: RhoA-GTP should be normalized to total RhoA and P-Akt should be normalized to total Akt. Tubulin cannot be used, especially since PF-8380 increases RhoA expression which means that the increase in RhoA-GTP is likely due to the increase in RhoA expression level.

4. Western blot quantification should be done consistently across figures and panels and in a way that makes experimental replicates comparable. This can be done by plotting the results as a percentage of the control as was done in Figure 7C left panel, but not the right panel. It would be helpful to include how the quantification was done in the methods section.

5. The RhoA activity assay method (Figure 7A and S7B) should be included in the methods section.

Minor points:

1. The spacing and formatting of the figure legend titles is inconsistent.

2. Please define all abbreviations: MM, TAM, SCC, etc.

3. It would be beneficial to briefly introduce RhoA, p190RhoGAP, and p110 δ PI3K in the introduction.

4. It would be helpful to show the replicate number for each panel in the legend or, even better, by showing the replicates in the bar graphs as individual points.

5. Statistical analysis: Mann-Whitney is good when comparing 2 conditions. When 3 or more conditions are present (figure 6B-C), ANOVA is more appropriate.

Reviewer #3 (Remarks to the Author):

The manuscript entitled "A combined opposite targeting of p110 δ PI3K and RhoA abrogates skin cancer" describes a series of experiments using subcutaneous mouse models of skin cancers treated with siRNA against 190RhoGAP, or with macrophages from p110 δ PI3K mutant mice.

The authors find that combining p110 δ PI3K impaired macrophages with p190RhoGAP knockdown leads to diminished tumour growth, for melanoma (independent of BRAF V600E mutational status) or SCC. Next they find that treatment with p110 δ PI3K impaired macrophages reduces ATX levels in tumour sections. However, inhibition of ATX is insufficient to reduce tumour growth. The results would be of interest to the cancer field and companies supporting autotaxin inhibitor trials.

Overall the data shown supports the conclusions made, however, some additional experimental detail is required in places, and some data is alluded to but not shown. Overall the study would benefit from a better placement of the results relative to the existing literature. This is done in the discussion, however, it is also needed in the introduction. For example, why did the authors choose to disrupt RhoGAP and PI3K delta? What was the aim of the study – presumably to test whether PI3K delta inhibition would be a strategy to target skin cancer? Likewise, why did they go on to investigate ATX? With a significant rewrite the data have value to the field. I have focused my comments on improving the manuscript.

Major comments:

1) Please place the work in context. Why target RhoGAP and PI3K delta? Why investigate ATX? What is ATX (please include in the intro)? What is already known about PI3K delta mutant macrophages? How does siRhoGAP increase PTEN activity? Why not just use a pan PI3K inhibitor?

2) Please include n numbers for all experiments or show individual data points. E.g Figure 8B.

3) Please state from what stage tumour histology was performed – equivalent sizes or time post

injection? E.g. for Fig 6A

4) It is shown that ATX levels increase upon treatment with the ATX inhibitor (figure 8B), what do ATX levels look like in untreated tumours? They might also go up reflective of the increased TME. This is needed before concluding that the ATX inhibitor promotes feedback.

5) Ensure the meaning of error bars is stated in legend. E.g. missing from Fig S7 legend.

6) Please state the relevant statistical tests. What statistical test is used in Figure 2A? It compares multiple groups therefore cannot be Mann Whitney U test and should be a one way ANOVA.

7) It would be more informative in Figure 7 if the effect of the ROCK inhibitor and PTEN inhibitors alone were shown.

Minor comments.

1) A simple timeline figure would help with clarity for Fig 3, S3 and S4.

2) Page 6: Is there a reason that data is not shown for the patient tumour model?

3) Fig S1: Please describe the statistical test used and what *** represents in the legend.

4) Please correct to Mann Whitney U test.

5) Some Western blots have molecular weights labelled, others don't. Please include throughout.

6) Autotaxin is a secreted enzyme. For Fig S7C, p190RhoGap could be regulating its secretion, which is not assessed.

Reviewer #1

Major points:

- Introduction is too succinct; in particular, it is lacking important background on SCC and the proteins studied PI3K/PTEN, Rho and ATX; what is the meaning of the abbreviation SCC?

We have now included in “Introduction” section the background on SCC and on the proteins that are studied in the manuscript and we have also defined all abbreviations. The background on ATX is included in the “Discussion” section (page 10-11).

- Figure 1 To me, the macrophages are not equally distributed (around the tumours?); provide higher resolution and larger enlargements. Are the quantifications performed on the whole slide? Are there differences in pAKT activation in macrophages in both conditions?

- As suggested by the Referee we have now used the figure panels at high resolution (600 dpi) and we show magnified regions showing that when macrophages are recruited to tumour sites then are mainly equally distributed in the tumour environment.
- The quantifications performed on the whole slide.
- As suggested by the Referee we have isolated and analysed the TAMs fraction of the tumours (please see ‘Materials and Methods’ (page 16) and ‘Results’ section (page 5)) and we now show that the phosphorylation of Akt in TAMs fraction is affected under both conditions (Fig 1E and 1F of the revised manuscript).

-Provide Rho activity experiments

As suggested by the Referee we now additionally show RhoA activity experiments for B16 tumours (fig. S1B, “Results” section (page 5)) and 451Lu tumours (fig. S5B, “Results” section (page 7)). We had shown RhoA activity experiments also in the previous version of our manuscript for SCC tumours (fig. S7B, ‘Results’ section (page 8)).

- p6 show data on PDTXs models

As suggested by the Referee we now show the data from melanoma PDX experiments (Fig. 4D, “Results” section (page 7 of the revised version), ‘Materials and Methods’ (page 13)). We had not actually shown the results of those experiments because we had performed these studies in a limited number of human melanoma specimens.

- Demonstrate the role of ATX in the loop between tumoural Rho and macrophage’s PI3Kdelta

We have extensively studied the link among ATX, RhoA and p110delta in figures 6, S7 and 7 (‘Results’ section pages 8-9 and “Discussion” section page 11 of the revised version) and we show that the suppression of ATX expression under the combined opposite targeting of p110 δ PI3K and RhoA derives from both, the modulation of macrophage recruitment to tumour sites because of p110 δ PI3K inactivation, and the increased RhoA activity, because of the silencing of p190RhoGAP expression in tumour cells. We further show that the increased RhoA activity into tumours reduces the expression of ATX and on the other hand inhibition of ATX activity in cancer cells induces the expression and activity of RhoA indicating a negative reverse link between RhoA and ATX.

- Fig 4C + M&M: colony assay: how are cells grown ex vivo – do they maintain the same levels of activation of PTEN, Rho and ATX?

As suggested by the Referee we have now written in detail (page 14) the method of tumour blood burden measurement which is a direct evaluation of intravasation (Xue C, *et al. Cancer Research* **66**, 192-197 (2006)).

Concerning the activity of RhoA and PTEN in cells, we need to mention that a very high amount of cells are needed for reliable measurements and therefore the cancer cells that are eventually collected from the blood of mice are not in any case enough for those quantifications.

Concerning ATX activity, all methods that are available, at least in our knowledge, are applied in serum, plasma, cell culture media and biological fluids. Therefore it could not be feasible to measure ATX activity in cancer cells collected from the blood. However, we have determined the ATX activity in plasma and indeed found that ATX activity is reduced in the plasma of mice receiving p190RhoGAP siRNA and $\delta^{D910A/D910A}$ macrophages compared with mice receiving p190RhoGAP siRNA and WT macrophages (Figure S7E of the revised manuscript) indicating that the p190RhoGAP silencing and the inactivation of p110 δ affect also the secretion of ATX (please see also Referee's 3 minor comment '6').

- Discussion: recent publications on PI3K-dependent macrophage / tumour cell dialog in other cancer settings are missing in the discussion and could further argue for the importance of the results of the authors.

As suggested by the Referee we have now included published data showing the implication of PI3K in the interplay between TAMs and cancer cells in other cancer types ("Discussion" section, page 10)

Minor:

- Some scales are missing in some photos.

This has been rectified.

Reviewer #2

Major points:

1. Figures 2-6: It is not clear why the authors use p190RhoGAP siRNA without controls in the in vivo studies with the exception of Fig. 2A. It cannot be concluded that targeting p110 δ PI3K and p190RhoGAP has a combined effect on tumor growth if p190RhoGAP expression is silenced in all conditions. Non-targeting siRNA controls should be included if the authors are to conclude that the results are due to a combined effect of $\delta^{D910A/D910A}$ macrophages and silencing p190RhoGAP and not just the $\delta^{D910A/D910A}$ macrophages.

As suggested by the Referee we now show the tumour burden in mice receiving WT macrophages alone (and siRNA control) in all figure panels showing tumour volume (Fig. 2C and 2D, Fig. 4A and 4D, Fig. 5B).

We might also need to mention that the result under the experimental condition in which the mice received p190RhoGAP siRNA and WT macrophages shows only the outcome because of the p190RhoGAP silencing whereas the biological effects in mice receiving p190RhoGAP siRNA and

$\delta^{D910A/D910A}$ macrophages come from the combined effect of p190RhoGAP silencing and the inactivation of p110 δ . Therefore, the comparison of these two conditions is biologically relevant.

2. It cannot be claimed that the effects of silencing p190RhoGAP are RhoA-dependent without testing the impact of silencing p190RhoGAP on the activity of other Rho GTPases. While p190RhoGAP is the predominant RhoA GAP, it can also function as a GAP for Rac1 in cells and possibly others (for example, see <https://pubmed.ncbi.nlm.nih.gov/27481945/> and <https://pubmed.ncbi.nlm.nih.gov/23499677/>). Either additional experiments should be included to rule out that p190RhoGAP is acting on any other Rho GTPases in these cancer cell lines, or it should be noted that other Rho GTPases may be involved. Subsequently, the authors should state that they are targeting p190RhoGAP in the title and in the text, not RhoA, since RhoA exclusivity has not been conclusively shown.

As suggested by the Referee we have now also examined the activation of Rac1 upon p190RhoGAP silencing and we show that the levels of Rac1-GTP remain unaffected (fig. S1C, and ‘Results’ section (page 5)) indicating that p190RhoGAP affects selectively RhoA under these experimental conditions.

p190RhoGAP was found to act as GAP only on Rac1 in cells, apart from its predominant substrate which is RhoA. Therefore, we have not tested other Rho GTPases. We trust that the Referee will agree with us that functional studies on mice for other GTPases which have not proved substrates for p190RhoGAP in cells are beyond the scope of the current MS.

3. Figure 7A, middle panel and 7C, right panel: The effects on P-Akt: RhoA-GTP should be normalized to total RhoA and P-Akt should be normalized to total Akt. Tubulin cannot be used, especially since PF-8380 increases RhoA expression which means that the increase in RhoA-GTP is likely due to the increase in RhoA expression level.

As suggested by the Referee we have now normalized the results to total RhoA and total Akt respectively.

4. Western blot quantification should be done consistently across figures and panels and in a way that makes experimental replicates comparable. This can be done by plotting the results as a percentage of the control as was done in Figure 7C left panel, but not the right panel. It would be helpful to include how the quantification was done in the methods section.

As suggested by the Referee we now express the results as a percentage of the respective control.

5. The RhoA activity assay method (Figure 7A and S7B) should be included in the methods section.

This has been mistakenly removed and now has been included (‘Materials and Methods’ section (page 15)).

Minor points:

1. The spacing and formatting of the figure legend titles is inconsistent.

This has been rectified.

2. Please define all abbreviations: MM, TAM, SCC, etc.

We have now defined all abbreviations throughout the MS.

3. It would be beneficial to briefly introduce RhoA, p190RhoGAP, and p110 δ PI3K in the introduction.

We have now included in the “Introduction” section the background on the proteins that are studied in the manuscript.

4. It would be helpful to show the replicate number for each panel in the legend or, even better, by showing the replicates in the bar graphs as individual points.

This has been rectified.

5. Statistical analysis: Mann-Whitney is good when comparing 2 conditions. When 3 or more conditions are present (figure 6B-C), ANOVA is more appropriate.

We had actually used ANOVA for the comparison of the different conditions and then Mann-Whitney test to compare values per pair. However, we agree with the Referee and we now mention both tests in figure legends.

Reviewer #3

Major comments:

1) Please place the work in context. Why target RhoGAP and PI3K delta? Why investigate ATX? What is ATX (please include in the intro)? What is already known about PI3K delta mutant macrophages? How does siRhoGAP increase PTEN activity? Why not just use a pan PI3K inhibitor?

We have now included in the “Introduction” section a detailed background on the proteins that are studied in the manuscript. We had also included the background on ATX in the “Discussion” section (page 10-11).

2) Please include n numbers for all experiments or show individual data points. E.g Figure 8B.

This has been rectified.

3) Please state from what stage tumour histology was performed – equivalent sizes or time post injection? E.g. for Fig 6A

This information is now stated in the respective figure legends.

4) It is shown that ATX levels increase upon treatment with the ATX inhibitor (figure 8B), what do ATX levels look like in untreated tumours? They might also go up reflective of the increased TME. This is needed before concluding that the ATX inhibitor promotes feedback.

As suggested by the Referee we now show the ATX levels in untreated mice. The expression of ATX in cancer cells from harvested tumours of untreated mice was found to initially increase but

early during tumour progression remained at constantly low levels (Fig. 8 and “Results” section (page 9)).

5) Ensure the meaning of error bars is stated in legend. E.g. missing from Fig S7 legend.

This has been rectified.

6) Please state the relevant statistical tests. What statistical test is used in Figure 2A? It compares multiple groups therefore cannot be Mann Whitney U test and should be a one way ANOVA.

We had actually used ANOVA for the comparison of the different conditions and then Mann-Whitney test to compare values per pair. However, we agree with the Referee and we now mention both tests in figure legends.

7) It would be more informative in Figure 7 if the effect of the ROCK inhibitor and PTEN inhibitors alone were shown.

As suggested by the Referee we now show experiments including all different conditions (Fig. 7C).

Minor comments.

1) A simple timeline figure would help with clarity for Fig 3, S3 and S4.

As suggested by the Referee we now show a timeline figure panel on the mentioned figures.

2) Page 6: Is there a reason that data is not shown for the patient tumour model?

We had not actually shown the results of those experiments because we had performed these studies in a limited number of human melanoma specimens.

As suggested by all Referees we now show the data from melanoma PDX experiments (Fig. 4D, “Results” section (page 7 of the revised version), ‘Materials and Methods’ (page 13)).

3) Fig S1: Please describe the statistical test used and what * represents in the legend.**

This has been rectified.

4) Please correct to Mann Whitney U test.

This has been rectified.

5) Some Western blots have molecular weights labelled, others don't. Please include throughout.

This has been rectified.

6) Autotaxin is a secreted enzyme. For Fig S7C, p190RhoGap could be regulating its secretion, which is not assessed.

As suggested by the Referee we measured the activity of ATX in culture medium of p190RhoGAP siRNA transfected cells and found that it was reduced compared to that of mock transfected cells

(fig. S7D). We also measured the activity of ATX in the plasma of mice and we found that it was decreased in plasma from mice receiving p190RhoGAP siRNA and $\delta^{D910A/D910A}$ macrophages compared to that from mice receiving p190RhoGAP siRNA and WT macrophages (fig. S7E) indicating that the secretion of ATX is affected under both treatment conditions.

Reviewers' comments:

Reviewer #1 (Remarks to the Author):

The authors answers in a sufficient manner to my questions and substantially improved the quality of the results that substantiate their conclusions.

Reviewer #2 (Remarks to the Author):

The authors have addressed my concerns and the concerns of the other reviewers through additional experiments and revisions. In particular, my concerns regarding the tumor burden assays have been addressed through the inclusion of WT macrophages, as well as my concerns regarding RhoA specificity of p190RhoGAP have been sufficiently addressed through inclusion of a Rac1 activity assay. In my opinion, the manuscript is ready for publication.

Reviewer #3 (Remarks to the Author):

The authors have addressed many of the concerns raised regarding the original submission meaning that the data are now placed into context and more readily interpretable.

There remain a few issues to address arising from the corrections made:

1. Figure 2 legend: $p < 0.5$ should be 0.05?
2. Figure 7A. I notice that the normalization blot has been swapped out from Tubulin to RhoA in response to a reviewer comment, yet the pull down blot remains the same. I am confused by the quantification. The fold change over Tubulin was approximately 2.5 fold. However, despite an increase in RhoA levels in the treatment condition, the fold change remains approx 2.5 fold. Are the raw densitometry measurements available? Is the RhoA blot from the same lysate from which the pull downs were performed?
3. The figure 8 legend is missing therefore we cannot assess the n numbers used in the experiments.
4. Methods: What vector was used for shRNA lentivirus production? And where was it from? Currently: "lentiviral GFP plasmid vector".
5. Methods: Please include a paragraph detailing antibodies (where purchased from etc). I note some are included in the reporting summary but some are missing e.g. pAkt.

Reviewer #3:

The authors have addressed many of the concerns raised regarding the original submission meaning that the data are now placed into context and more readily interpretable.

There remain a few issues to address arising from the corrections made:

1. Figure 2 legend: p05 should be 0.05?

This is most likely a typographical error and now has been corrected.

2. Figure 7A. I notice that the normalization blot has been swapped out from Tubulin to RhoA in response to a reviewer comment, yet the pull down blot remains the same. I am confused by the quantification. The fold change over Tubulin was approximately 2.5 fold. However, despite an increase in RhoA levels in the treatment condition, the fold change remains approx 2.5 fold. Are the raw densitometry measurements available? Is the RhoA blot from the same lysate from which the pull downs were performed?

The blot showing total RhoA is from the same lysates that were used for the pull down of RhoA-GTP. The fact that the fold change remained the same seems weird but it is actually reasonable because the increase in RhoA-GTP levels in the treatment condition compared to untreated cells is much higher than the increase in total RhoA levels in PF-8380-treated compared to untreated cells. This is also depicted on the raw data from 3 representative experiments that presented below:

Exp.1

	RhoA-GTP	Total RhoA	RhoA-GTP/T-RhoA	RhoA-GTP/T-RhoA (% of control)
vehicle	82450	175296	0.470347298	100
PF-8380	356740	299229	1.192200552	253.4723928

Exp.2

	RhoA-GTP	Total RhoA	RhoA-GTP/T-RhoA	RhoA-GTP/T-RhoA (% of control)
vehicle	63544	153457	0.414083424	100
PF-8380	228758.4	257858	0.887148741	214.2439637

Exp.3

	RhoA-GTP	Total RhoA	RhoA-GTP/T-RhoA	RhoA-GTP/T-RhoA (% of control)
vehicle	97427	195669	0.497917401	100
PF-8380	448164	293720	1.525820509	306.440487

3. The figure 8 legend is missing therefore we cannot assess the n numbers used in the experiments.

The legend for figure 8 was included in our manuscript (page 22 of the previous version, page 23 of the current version) mentioning also the 'n' numbers used in the experiments.

4. Methods: What vector was used for shRNA lentivirus production? And where was it from? Currently: "lentiviral GFP plasmid vector".

The ARHGAP15 human, 4 unique 29mer shRNA constructs in pGFP-C-shLenti vector were purchased from Origene (cat no TR312603). This information is now included in the Methods section (page 17).

5. Methods: Please include a paragraph detailing antibodies (where purchased from etc). I note some are included in the reporting summary but some are missing e.g. pAkt.

As suggested by the Referee we have now included a paragraph (page 12) including details for all antibodies used in the study.